# Dimeric G-quadruplex motifs-induced NFRs determine strong replication origins in vertebrates

Jérémy Poulet-Benedetti[1,8], Caroline Tonnerre-Doncarli[1,8], Anne-Laure Valton [1,6,7], Marc Laurent[1], Marie Gérard[1], Natalja Barinova[1], Nikolaos Parisis[1], Florian Massip [2,3,4], Franck Picard [5] ✉ & Marie-Noëlle Prioleau [1] ✉

Replication of vertebrate genomes is tightly regulated to ensure accurate duplication, but our understanding of the interplay between genetic and epigenetic factors in this regulation remains incomplete. Here, we investigated the involvement of three elements enriched at gene promoters and replication origins: guanine-rich motifs potentially forming G-quadruplexes (pG4s), nucleosome-free regions (NFRs), and the histone variant H2A.Z, in the firing of origins of replication in vertebrates. We show that two pG4s on the same DNA strand (dimeric pG4s) are sufficient to induce the assembly of an efficient minimal replication origin without inducing transcription in avian DT40 cells. Dimeric pG4s in replication origins are associated with formation of an NFR next to precisely-positioned nucleosomes enriched in H2A.Z on this minimal origin and genome-wide. Thus, our data suggest that dimeric pG4s are important for the organization and duplication of vertebrate genomes. It supports the hypothesis that a nucleosome close to an NFR is a shared signal for the formation of replication origins in eukaryotes.

DNA replication is a tightly constrained and highly regulated process that allows cells to completely duplicate their genome during each cell cycle. Replication is initiated at sequences called origins of replication and occurs in units of about 0.2–2 Mb (replication timing domains) at early, mid, or late replication[1–3]. The spatiotemporal regulation of this program is controlled by complex interactions between multiple genetic and epigenetic determinants that are bound by essential initiation proteins forming the pre-replication complex (pre-RC), at the origin of replication. In budding yeast, the genetic determinant is an 11-bp consensus sequence motif that defines an autonomous replication sequence, and additional neighboring *cis*-elements are needed to form

a nucleosome-free region (NFR) and an effective origin[4]. A recent study, imaging budding yeast ORC's (origin recognition complex) behavior on nucleosomal DNA in real time, revealed that the replicative helicase MCM (mini-chromosome maintenance) double hexamers loading depends on the presence of a nucleosome next to an NFR and does not require a high consensus motif[5]. The authors hypothesized that nucleosomes next to an NFR are primarily responsible for origin function in all cells. In vertebrates, genome-wide mapping of replication origins has revealed a strong association of efficient origins with CpG islands (CGIs) and promoters[6,7]. Rather than a consensus sequence as seen in yeast, however, the only vertebrate determinant

[1]Université Paris Cité, CNRS, Institut Jacques Monod, F-75013 Paris, France. [2]MINES ParisTech, PSL-Research University, CBIO-Centre for Computational Biology, 75006 Paris, France. [3]Institut Curie, Paris, Cedex, France. [4]INSERM, U900 Paris, Cedex, France. [5]Laboratory of Biology and Modelling of the Cell, Université de Lyon, Ecole Normale Supérieure de Lyon, CNRS, UMR5239, Université Claude Bernard Lyon 1, Lyon, France. [6]Present address: Department of Systems Biology, University of Massachusetts Chan Medical School, Worcester, MA, USA. [7]Present address: Howard Hughes Medical Institute, Chevy Chase, MD, USA. [8]These authors contributed equally: Jérémy Poulet-Benedetti, Caroline Tonnerre-Doncarli. ✉e-mail: franck.picard@ens-lyon.fr; marie-noelle.prioleau@ijm.fr

identified is contained in a cage-like local secondary DNA structure of guanine-rich sequences known as potentially forming G-quadruplexes (pG4s). These motifs are enriched at replication origins and are essential for origin activity[8–11]. However, a single pG4 is insufficient to define an active origin, suggesting that a more complex assembly of elements could be necessary to trigger replication initiation[9,12]. As in yeast, strong replication origins in vertebrates co-localize with NFRs and are associated with epigenetic marks such as the histone variant H2A.Z, which has been shown to promote origin activity[13–17]. Despite recent advances in understanding the key regulatory elements of replication initiation, the lack of a consensus operational *cis*-regulatory element(s) driving origin formation has thus far prevented a causal link to be established between *cis*-elements and specific chromatin organization at origins.

In this study, we investigated the requirements for and the interplay between pG4s, NFRs, and H2A.Z in the formation of origins of replication using the chicken vertebrate β^A-globin promoter/origin as a model system. Previous studies have shown that this origin was active in different chromatin contexts including a late region strongly associated with lamin B1[18]. In addition, a series of point mutations in a canonical pG4 showed a close correlation between the activity of this origin and the stability of the structured G4, suggesting its involvement in origin function[9]. However, we showed that this G-quadruplex had to cooperate with another region covering 250 bp to form an efficient origin. In this new study, we sought to identify which other *cis*-element was necessary and sufficient to form an efficient origin with the intention of identifying the constraints acting on the replication initiation machinery. Using a similar genetic approach, we show here that same-strand dimeric pG4s are necessary and sufficient for the initiation of replication within the model origin, and that this activity is correlated with the formation of an NFR next to well-positioned nucleosomes. We also demonstrate that dimeric pG4s are associated with specific NFR formation patterns at replication start sites genome-wide. Our results identify dimeric pG4s as a new functional genetic element potentially involved in the formation and regulation of ~ 36% of strong replication origins in human and chicken. Their action is linked, among other things, to their ability to organize NFRs next to well-positioned nucleosomes. This strategy is in line with the recognition mode of the ORC complex in eukaryotes.

## Results

### The presence of dimeric pG4s is essential for the chicken β^A-globin origin activity

To delineate novel *cis*-motifs essential for metazoan replication origin activity within gene promoters, we employed a model origin construct that has the capacity to induce strong enrichment of short nascent strands (SNSs), a characteristic marker of replication origins, and to locally advance the replication timing (RT) of a middle-late replicated region in population-based assays in avian DT40 cell lines (Fig. 1a–c). We used the 1.1-kb chicken β^A-globin promoter/origin associated with the IL2R reporter gene fused to a poly-A sequence and flanked by two upstream stimulatory factor (USF) binding sites (Fig. 1a). This wild-type construct and mutant versions with specific elements deleted were inserted at the same locus by homologous recombination. The wild-type construct did neither induce transcription in the reporter gene or in the promoter sequence (Supplementary Fig. 1). We previously identified the essential role in the activity of this model origin of a pG4 (pG4#1 in Fig. 1a) and the downstream 245-bp sequence containing an erythroid-specific factor binding site (CACC box) as well as CCAAT and TATA boxes, which are binding sites for the ubiquitous transcription factors nuclear factor Y and TATA-binding protein, respectively (Fig. 1a)[9]. We quantified the capacity of each mutated origin to initiate replication by measuring SNS enrichment in two independent clones. We found that single or combined deletion of the CCAAT and TATA boxes slightly decreased the relative SNS enrichment compared with

the full β^A-globin origin, indicating a limited but detectable impact on origin activity (Fig. 1c). Focusing on the CACC box and the partially overlapping pG4#2 (located on the complementary strand of pG4#1), deletion of the box and the adjacent downstream 12-bp sequence of pG4#2 (ΔCACC + ΔpG4#2 mutant) slightly increased origin activity (Fig. 1c). In contrast, deletion of the region containing a third pG4 (pG4#3) framed by the CCAAT and the TATA boxes (Δ(CCAAT to TATA) mutant) resulted in a dramatic loss of replication activity, comparable to that resulting from deletion of the entire 245-bp module (Fig. 1c)[9]. To confirm the essential role of pG4#3, we deleted CCAAT and TATA boxes and replaced three guanines with adenines to eliminate the three triplets of Gs (Fig. 1a, asterisks). These mutations resulted in a degree of SNS enrichment similar to that observed when the entire CCAAT box to TATA box sequence was deleted (Fig. 1c, compare Δ(CCAAT to TATA) with ΔCCAAT + ΔTATA mpG4. A series of point mutations in pG4#1 performed previously showed a strong correlation between the stability of the G4-quadruplex structure and the efficiency of the origin, suggesting the involvement of the structuring of this pG4 in the origin activity. Our study does not allow us to conclude whether pG4#3 should also keep its ability to structure itself, but we can conclude that both pG4#1 and pG4#3 are critical for β^A-globin origin activity.

### The TATA box plays a role in RT control

While quantification of SNS enrichment at the inserted β^A-globin promoter/origin assesses its capacity to induce replication initiation, measuring RT shifts can establish whether a change in initiation is occurring in the majority of a cell population, and also identifies an advance or delay in RT as positive or negative profile shifts, respectively[9,18,19]. We previously described a method to quantify the RT shift by measuring proportion differences between early and late fractions[9,18,20]. The RT shift observed when the full β^A-globin promoter/origin is inserted in the genome on both chromosomes points to the presence of *cis*-elements that ensure early initiation of this origin in the majority of cells (Fig. 1b). As shown in previous studies, the active β^A-globin origin (100% relative SNS enrichment) showed a significant median RT shift of +20.2, which compared with a median shift of +6.5 (~ 10% relative SNS enrichment) for inefficient origins (Fig. 1d)[9,18]. Insertion of the mutant origins that were found to have little to no effect on origin activity (see Fig. 1c) resulted in a positive RT shift for ΔCCAAT and ΔCACC + ΔpG4#2 and a negative shift for ΔTATA and ΔCCAAT + ΔTATA box origins (Fig. 1d and Supplementary Figs. 2 and 3 and 4A). In contrast, insertion of the inactivated Δ(CCAAT to TATA) and ΔCCAAT + ΔTATA mpG4 mutant origins, in which pG4#3 is disabled, had no effect on RT (Fig. 1d and Supplementary Figs. 4B and 5). These results identify a prominent role of the TATA box on RT. Notably, the delay in RT observed when the TATA box was deleted in active (but not inactive) origins suggests that the origin affects the initiation of flanking origins and/or replication fork progression in the altered region; however, the mechanism by which this may occur is beyond the scope of this study and must await further investigation. Overall, these results confirm the results obtained with relative SNS enrichment and indicate that the TATA box in the β^A-globin promoter/origin DNA region is a key *cis*-acting element for RT control, together with the previously identified USF binding sites[20].

### Identification of an active metazoan autonomous minimal replication origin

To determine whether particular combinations of pG4#1, pG4#3, CCAAT box, and/or TATA box are necessary and sufficient to induce the formation of a functional and efficient origin, we designed a minimal 90-bp β^A-globin origin containing pG4#1 followed by a 13-bp linker naturally present in the origin sequence, and the CCAAT to TATA box sequence containing pG4#3 (Fig. 2a). This minimal origin resulted in a 2.5 increase in SNS relative enrichment compared with the active

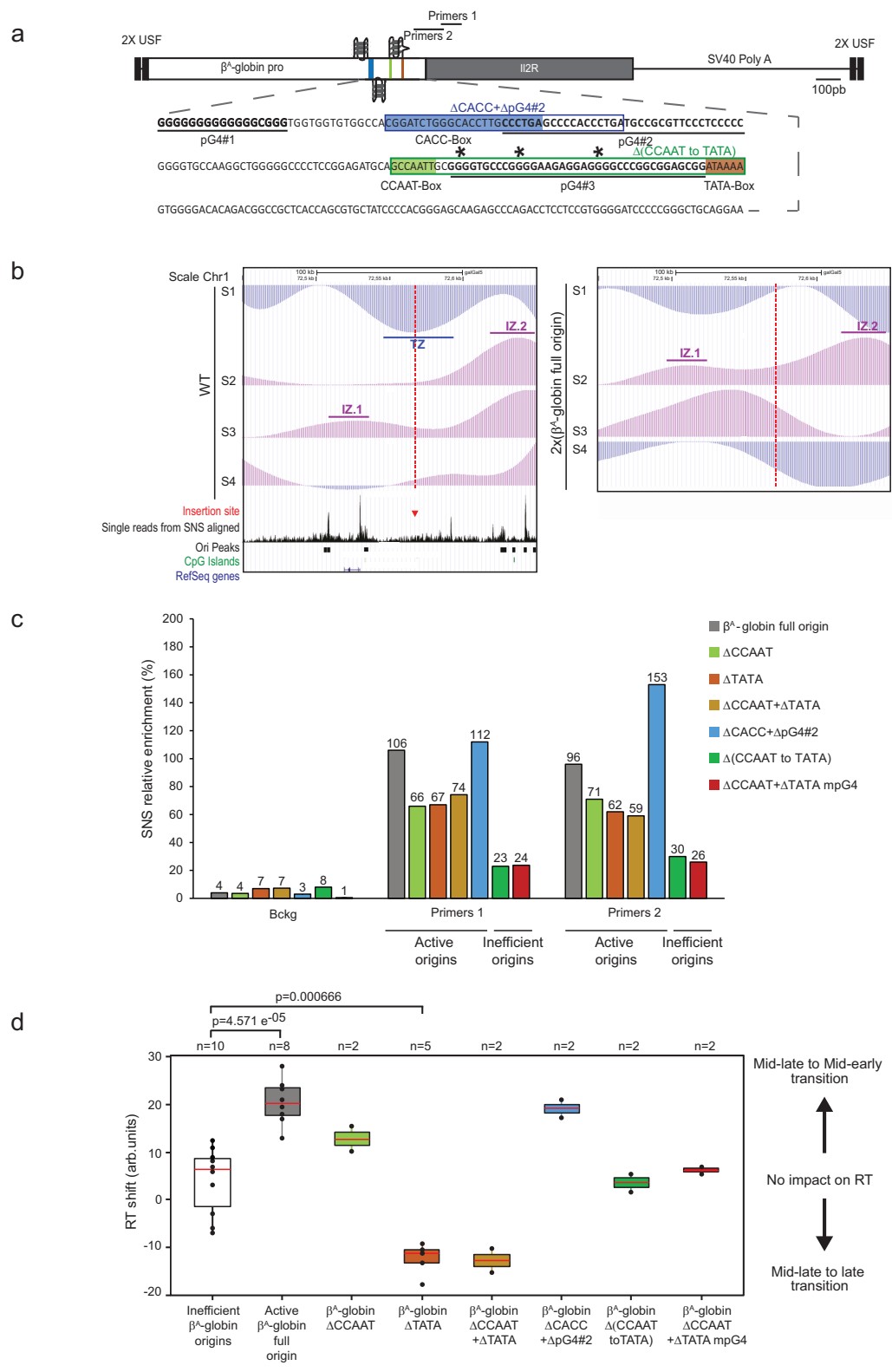

β<sup>A</sup>-globin full origin (Fig. 2b). This increased activity may have resulted from the shift in primer pairs used for SNS quantification relative to the pG4s in the minimal origin compared with the full origin (152 bp vs 325 bp downstream of pG4#3). In agreement with this hypothesis, a new primer pair (primers 1′), located 337 bp downstream of pG4#3 in the minimal origin, resulted in an SNS enrichment of 125%, similar to that observed for the full origin (Fig. 2b). In addition, the minimal

origin gave a median RT shift of +15.2, which is also comparable to that of the active full origin (Fig. 2c and Supplementary Fig. 6). These results suggest that the 90-bp minimal β<sup>A</sup>-globin origin is active and efficient and that these *cis*-elements are sufficient to form a functional origin. To probe this further, we deleted the CCAAT and TATA boxes from the minimal origin (Fig. 2a, middle) and examined the effects on SNS enrichment and RT shift. Notably, these results demonstrated that

**Fig. 1 | Identification of two pG4s essential for replication origin activity.**
**a** Structure of the ectopic chicken β^A-globin full origin. pG4#1, pG4#2, and pG4#3 are bold underlined; CACC box is in blue; CCAAT box is in green; and TATA box is in brown. Boxed outlines represent the large deletions in the 'ΔCACC + ΔpG4#2' and 'Δ(CCAAT to TATA)' mutant origins. Asterisks represent the three guanines mutated to alanines in ΔCCAAT + ΔTATA mpG4. Black lines below and above Primer pairs 1 and 2 indicate the position of amplicons used for short nascent strand (SNS) quantification. **b** BrdU pulse-labeled cells were sorted into four S-phase fractions from early to late (S1 to S4) and the immune-precipitated newly synthesized strands (NS) were deep sequenced. RT profiles observed at the targeted middle-late insertion region of chromosome 1 (genomic position: chr1: 72,450,000–72,650,000 bp; 200 kb; galGal5). The insertion site is indicated with a red dotted line. Tracks of nascent strands (NS) enrichments in the four S-phase fractions are shown separately (S1–S4) for the WT and a cell line harboring the ectopic wild-type β^A-globin full origin at the two chromosomes 2 × (β^A-globin full origin) (right). NS-enriched and depleted regions for each fraction are represented in purple and blue, respectively. Initiation zones (IZ) and termination zones (TZ) are labeled. Single reads from SNS-aligned, tracks of replication origins (Ori peaks) determined in ref. 35, annotated (RefSeq) genes, and CpG islands are shown below. **c** Relative SNS enrichment for the β^A-globin full origin and the indicated mutants (*n* = 2), mean values are indicated above each bar. Primers 1 and 2 refer to the two amplicons at the initiation site of the ectopic origin and Bckg refers to the amplicon located 5 kb away from the site of integration (background signal), respectively. One amplicon within the endogenous ρ-origin was arbitrarily set at 100% to quantify the relative SNS abundance. **d** Distribution of RT shift values for clonal cell lines containing active or mutant origins. Boxes show the median (red line) and 0.25 and 0.75 quartiles (lower and upper box edges), and minimum and maximum RT shift values (whiskers). Filled circles represent individual clones (*n* = 2–10 as indicated). *P* values were obtained using Wilcoxon's nonparametric two-tailed test. Source data are provided as a Source Data file.

the CCAAT and TATA boxes are not essential for the replication initiation activity of the minimal origin but deletion of the TATA box impairs the ability of the minimal origin to locally advance the RT (Fig. 2b, c and Supplementary Fig. 7).

## The replication origin determinant is defined by two pG4s on the same DNA strand

We previously showed that the orientation of pG4#1 relative to the 245-bp ectopic β^A-globin module determined the location of replication initiation 3′ of the G tract found inside pG4#1 (Fig. 2d and ref. 9). Therefore, to refine our definition of a functional replication origin, we investigated the effect of switching pG4#1 to the reverse complementary strand on the activity of the β^A-globin minimal origin (pG4#1 rev compl, Fig. 2a, bottom). Surprisingly, this switch eliminated SNS enrichment both downstream and upstream of pG4#1 (primer pairs 0 and 1, Fig. 2b), suggesting that the positioning of pG4#1 and pG4#3 on the same strand is essential for minimal origin activity. This raised the question of why pG4#1 could be inverted when associated with the 245-bp module (Fig. 2d). We hypothesized that pG4#2 may be able to cooperate with pG4#1 rev compl as part of a functional origin. To test this hypothesis, we employed a previously reported pG4#1 deletion β^A-globin origin (ΔpG4#1), which is devoid of origin activity[9], and attempted to rescue the wild-type origin activity by switching pG4#2 to the same strand as pG4#3 (ΔpG4#1-pG4#2 compl, Fig. 3a). Consistent with our hypothesis, we observed a diffuse enrichment of SNS (~30% at positions 1, 2, and 1′, Fig. 3b) and also detected a significant RT shift toward earlier replication (Fig. 3c and Supplementary Fig. 8) with the ΔpG4#1-pG4#2 compl origin. The latter result confirmed that this origin is active and also suggested that the pattern of SNS enrichment observed probably reflected a more diffuse replication initiation site. Taken together, these results indicate that the firing of this model origin depends on the presence of two pG4s positioned on the same strand. We found that all three possible pairs of pG4 combinations led to a functional origin, as did the presence of linkers of various sizes, suggesting that there is considerable flexibility in the arrangement of pG4s within the model origin (Fig. 3d). Conversely, no combination tested where two pG4s were on opposite strands worked (Fig. 3d).

## Chromatin organization at the minimal β^A-globin origin is similar to that observed at strong origins

In vertebrates, "strong" replication origins (i.e., highly efficient origins, experimentally defined as the top 25% most active origins in terms of SNS enrichment) are characterized by three important features. First, initiation occurs at a well-positioned but labile nucleosome; second, they are associated with an NFR encompassing the G-rich sequence; and third, the histone variant H2A.Z is associated with nearby nucleosomes[13,15,21]. However, because most efficient origins are found at active transcription start sites (TSSs), it is difficult to dissociate the

effects of transcription from those of the replication process on nucleosome organization. Our β^A-globin minimal origin circumvents this problem because it does not drive transcription and therefore allows us to examine only the causal link between nucleosome organization and replication origin function (Supplementary Fig. 1). We used micrococcal nuclease–deep sequencing (MNase-Seq) to map nucleosomes on the minimal origin and included the inactive pG4#1 rev compl minimal origin as a negative control. The minimal and mutant origins were inserted on both chromosomes in DT40 cells and MNase-Seq was performed in synchronized cells at the G1/S transition (pre-RC-bound), synchronized cells in G2 phase (not pre-RC-bound) or in asynchronous cells (Supplementary Figs. 9 and 10). We observed well-positioned nucleosomes (nucleosomes +1 to +5, Fig. 4a) flanking an NFR at the β^A-globin minimal origin. Nucleosomes +2 and +3 overlap and therefore cannot coexist on the same chromosome. Upstream of the minimal origin (and the large NFR), nucleosome organization is less well defined but at least four discrete positions can be delineated. Finally, we found that the large NFR included the 90-bp construct and about 100 additional bp upstream, leaving the 2X USF binding sites uncovered. In contrast to the functional β^A-globin minimal origin, no NFR was detected at the 90-bp minimal non-functional origin pG4#1 rev compl (Fig. 4b). Instead, a well-positioned nucleosome could be detected, suggesting that the presence of two pG4s alone is insufficient to exclude the nucleosome. In addition, a merged nucleosome (Fig. 4b, +2/3) replaced the discrete +2 and +3 nucleosomes detected in the active minimal origin. Overall, the positioning of nucleosomes observed on and around the active β^A-globin minimal origin resembled the pattern found genome-wide on aggregated origins containing a G tract[15]. To determine whether H2A.Z was enriched at these origins, we performed H2A.Z chromatin immunoprecipitation (ChIP) assays and we found enrichment of the histone variant on both the active and inactive origins predominantly at nucleosomes +1 to +5 (Fig. 4), suggesting that pG4 density controls H2A.Z recruitment independently of pG4 orientation. Thus, the precise nucleosome organization, including the NFR, depends on the presence of two same-strand pG4s (dimeric pG4s), and the results clearly demonstrate that a single pG4 is insufficient to organize a nucleosome pattern characteristic of strong origins. Moreover, we can conclude that the transcriptional machinery is not involved in establishing this nucleosome pattern, given that this minimal origin is not transcribed (Supplementary Fig. 1). Finally, we observed that the NFR is already formed in G2, a time when the pre-RCs are not loaded, suggesting that licensing is not responsible for its formation (Fig. 4a).

## The med14 promoter/origin is also under the control of two same-strand pG4s

We had previously investigated the role of pG4s in origin function of the med14 promoter region[9]. We had identified five pG4s downstream of the origin on the minus strand (pG4#1 to 5, Fig. 5a, b) and a recent

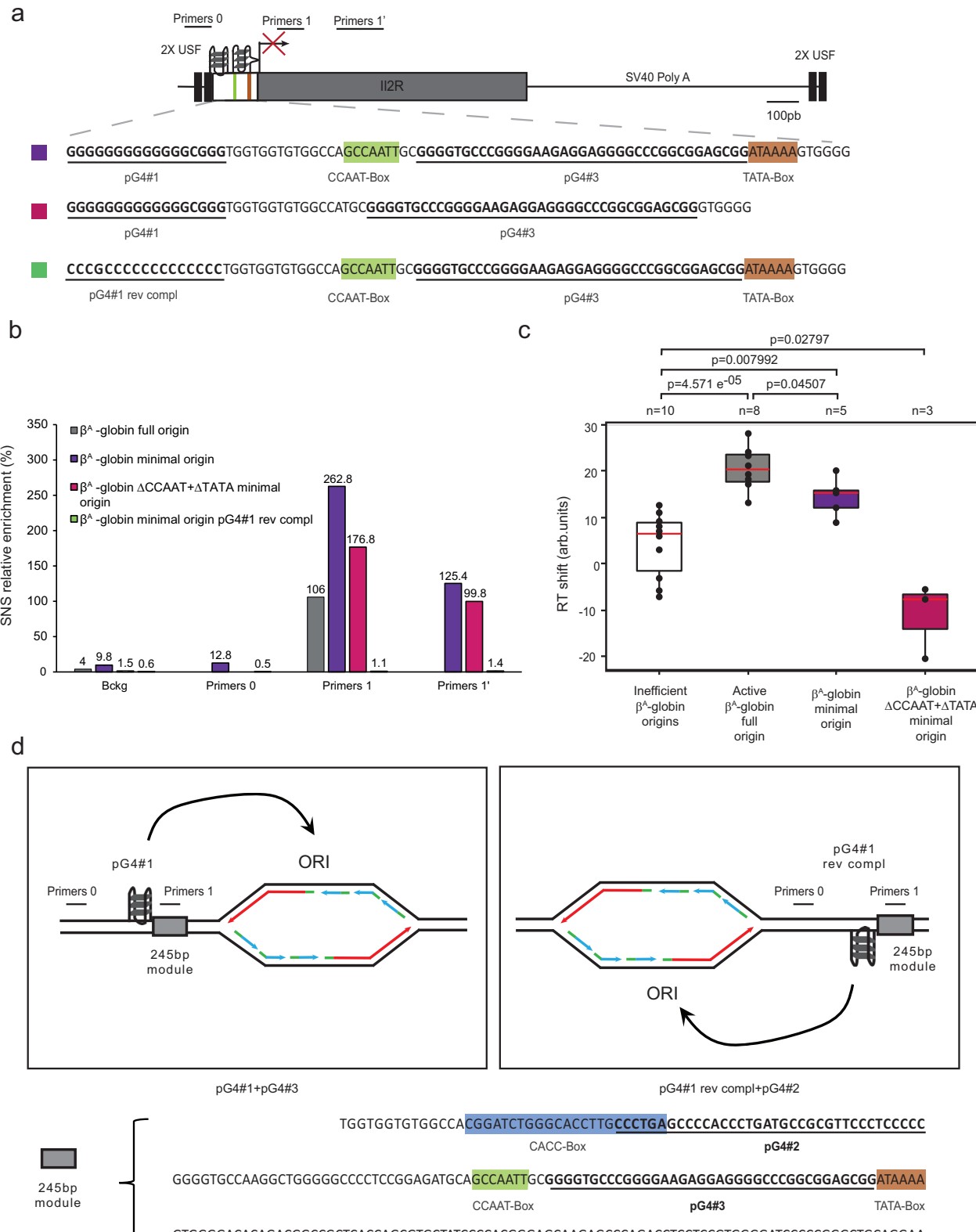

**Fig. 2 | Two pG4s on the same DNA strand are sufficient to form an efficient origin. a** Structure of the ectopic β^A-globin minimal origin. Mutant sequences with CCAAT and TATA boxes deleted (ΔCCAAT + ΔTATA), and pG4#1 switched to the reverse complement strand (pG4#1 rev compl) are shown. See Fig. 1a for the explanation of colored and underlined sequences. Positions of primers used for SNS enrichment are indicated at the top. **b** Relative SNS enrichment for the β^A-globin full or minimal origin and mutants (n = 2), mean values are indicated above

each bar. **c** Distribution of RT shift values for the two mutant containing active origins. Boxes are as described for Fig. 1d (n = 3–10 as indicated). P values were obtained using a Wilcoxon's nonparametric two-tailed test. Source data are provided as a Source Data file. **d** Schematic showing the impact of pG4#1 orientation with respect to the 245 bp module (sequence shown below) on the location of the origin previously observed in (9). The pG4s hypothetically involved in the initiation of replication are indicated on the bottom of each box.

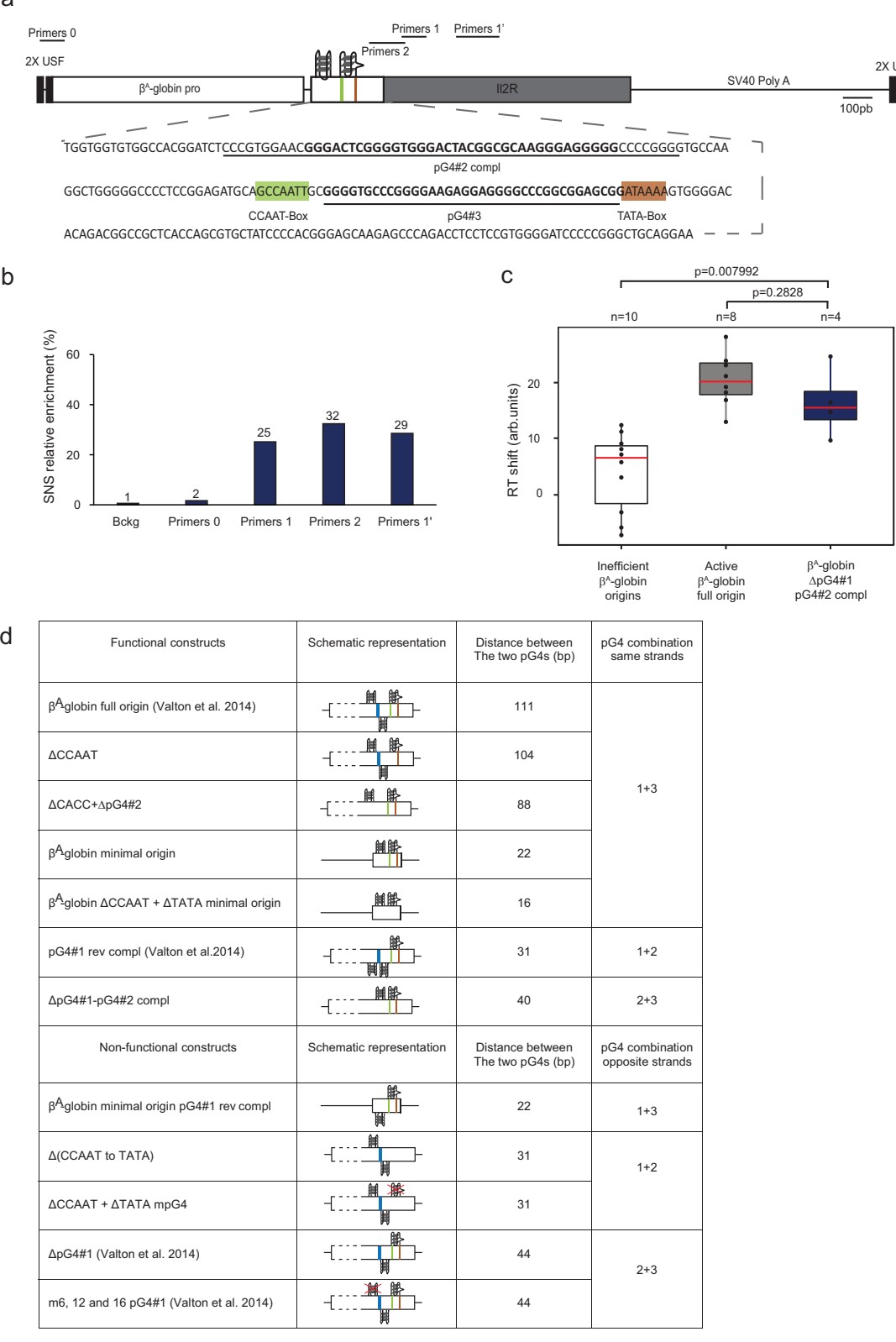

**Fig. 3 | The combination of dimeric pG4s necessary to form an efficient origin is flexible. a** Structure of the ectopic β^A-globin ΔpG4#1-pG4#2 compl origin. See Fig. 1a for the explanation of shaded and bold sequences. Positions of primers used for SNS enrichment are indicated at the top. **b** Relative SNS enrichment for the ΔpG4#1-pG4#2 compl mutant (*n* = 2), mean values are indicated above each bar. **c** Distribution of RT shift values for the ΔpG4#1-pG4#2 compl clonal cell line. Boxes are as described for Fig. 1d (*n* = 4–10 as indicated). *P* values were obtained using a Wilcoxon's nonparametric two-tailed test. Source data are provided as a Source Data file. **d** List of functional and non-functional β^A-globin origin constructs. Schematic representations of β^A-globin constructs indicate the orientation of pG4s (pG4s inactivated by point mutations are covered by a red cross). The CACC box is in blue; the CCAAT box is in green; and the TATA box is in brown. The distance in bp between the end of the first pG4 and the beginning of the second pG4 on the same strand (functional constructs) or opposite strand (non-functional constructs) and the specific pG4s combinations involved in their formation are indicated.

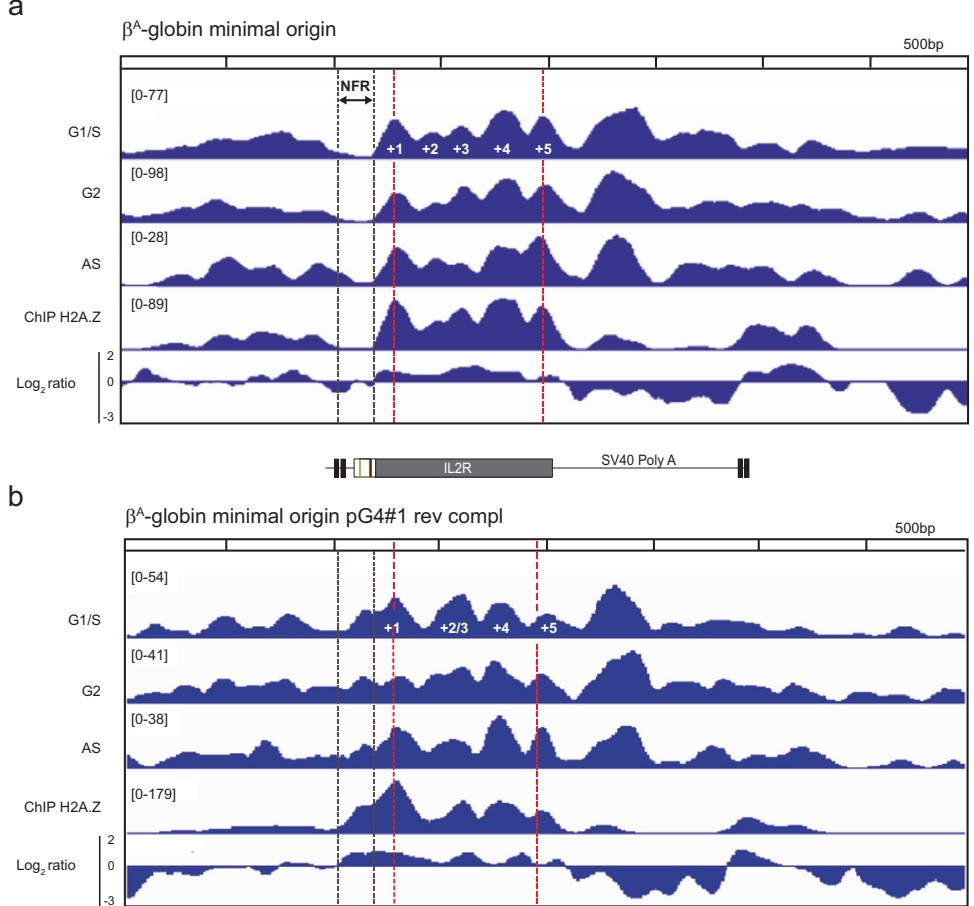

**Fig. 4 | Chromatin organization at the β^A-globin minimal origin is similar to that at strong origins. a** Nucleosome occupancy profiles (by MNase-Seq) along the ectopic β^A-globin minimal origin and in flanking regions, measured in synchronized cells at the G1/S transition and G2 and in asynchronous cells (AS). The lower two profiles show ChIP results with anti-H2A.Z antibodies on AS samples and the Log2 ratio between ChIP H2A.Z and MNase-Seq data. **b** Analysis as in (**a**) for the inactive pG4#1 rev compl minimal origin. A schematic of the construct is shown between (**a**) and (**b**). Black dotted lines delineate the NFR, and red dotted lines indicate the position of the centers of nucleosomes +1 and +5.

pG4 annotation revealed a sixth pG4 (pG4#Un, Fig. 5b)[22]. We had tested the role of pG4#4 and 5 because they were located at about 300 bp from the center of the origin peak, corresponding then to the average distance between the pG4s and the origin center (Fig. 5a). We had shown that in situ deletion of either one maintained significant origin activity while simultaneous deletion of pG4s#4 and 5 abolished the origin and concluded that the two pG4s cooperated to drive efficient initiation[9]. This result suggests a similar mode of operating to that observed for the β^A-globin origin where two pG4s on the same strand would be the key signal to ensure efficient origin formation. However, we noticed that although the combinations, 4 and 5 (WT), 3 and 5 (ΔpG4#4) as well as 3 and 4 (ΔpG4#5) were functional, the combination 2 and 3 (ΔpG4#4 and 5) did not work. This last result suggests that not all the identified pG4s have the same properties and is in agreement with our genome-wide analysis presented in the next paragraph, showing that in human and chicken, only 50% of clustered pG4s are associated with replication origins (Fig. 6b). The chromatin organization of the med14 origin is similar to that of the β^A-globin origin, with a very pronounced NFR covering pG4s#3 to 5 next to well-positioned nucleosomes enriched in histone variant H2A.Z (Fig. 5a). To determine whether the pG4s#4 and 5 found within the med14 promoter/origin are sufficient to form an origin of replication, we inserted two distinct constructs containing pG4s#4 and 5 at the same chromosomal site as the β^A-globin origin (Fig. 5c). Insertion of both constructs induced a high level of SNS relative enrichment showing again flexibility in the distance between the two pG4s (72 and 26 bp for Med14 pG4#4 + 5 and

Med14 pG4#4 + 5 minimal origin respectively) (Fig. 5c). These results confirm on a second model origin that two pG4s on the same strand is a key signal to induce the formation of an origin of replication.

## Replication origins are associated with clustered pG4s genome-wide

To evaluate whether same-strand dimeric pG4s are required to trigger the formation of an NFR and to fire origins, we investigated their colocalization genome-wide. Current estimates suggest that the human, chicken and mouse genomes contain ~1.2 million, ~350,000 and ~1.2 million pG4s, respectively (representing 1.41%, 1.36% and 1.5% of the genome sizes), among which about 85% are monomeric, ~13% are in cluster (defined here as at least two pG4s and up to 6 pG4s < 100 bp apart), (Fig. 6a)[22]. Clustered pG4s will refer to all pG4s grouped in pairs or more. In human, chicken and mouse cell lines, we found that clustered pG4s were more commonly associated with functional elements (CGIs, TSSs, origins), H2A.Z, and NFRs genome-wide compared with monomeric pG4s (Fig. 6b). In particular, 47% of clustered pG4s were associated with replication origins in human, 52% in chicken and 35% in mouse compared with ~30% of monomeric pG4s (27% in human, 31% in chicken and 26% in mouse). This trend was observed with core origins[23] and origins mapped with the Ini-seq2 method[16] (Supplementary Table 1). Structured G4s have recently been mapped in vivo in human, chicken and mouse cells by using artificial G4 probe proteins that binds structured G4s with high affinity and specificity[22]. We also found that clustered pG4s were more associated with detected structured G4

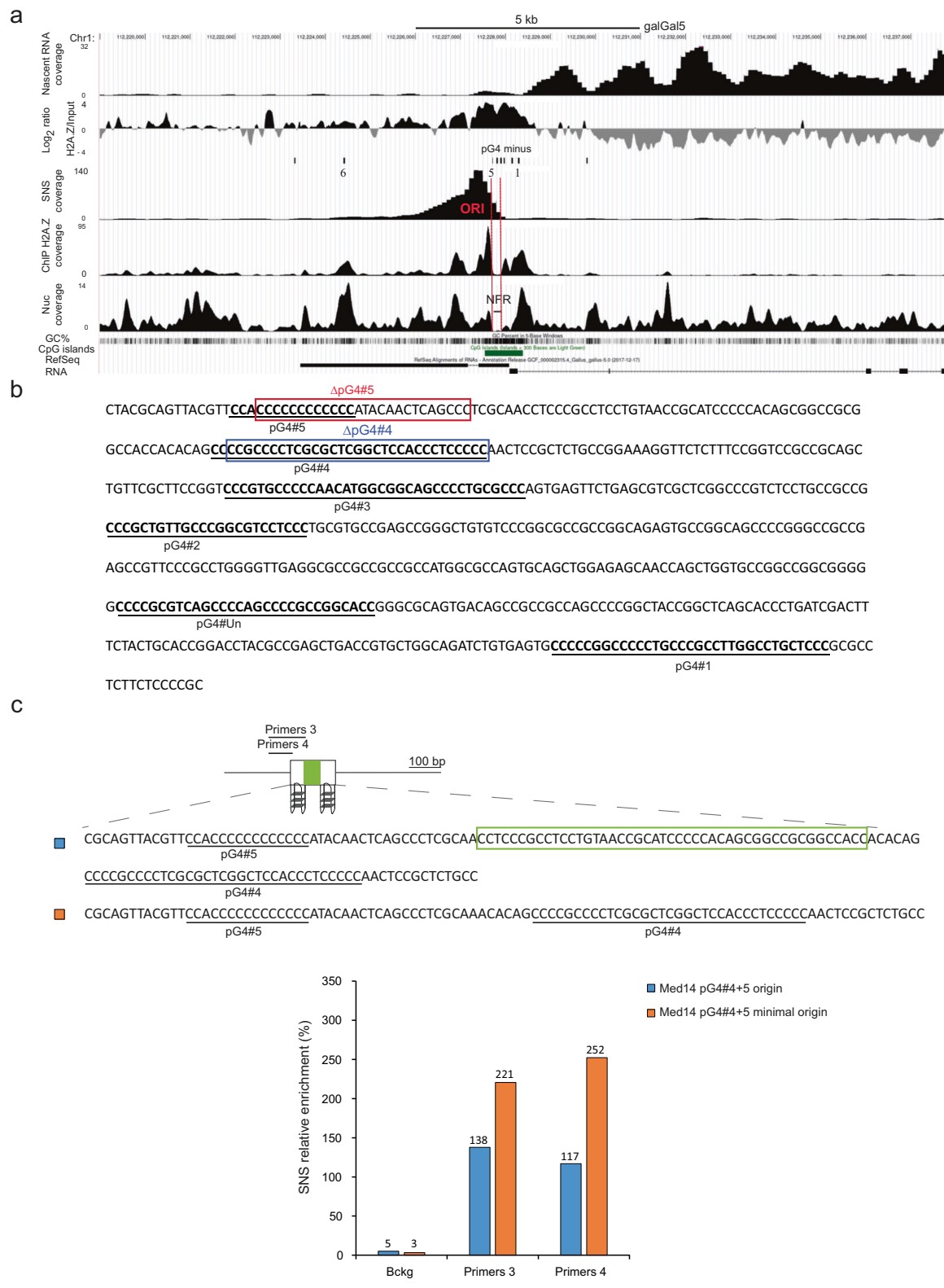

**Fig. 5 | Two same-strand pG4s determines the Med14 promoter/origin.**
**a** Genome browser profiles around the Med14 promoter/origin (genomic position: chr1: 112,219,000–112,238,000 bp; 200 kb; galGal5). Nascent RNA coverage, Log2 ratio between ChIP H2A.Z and MNase-Seq data, pG4s annotation on the minus strands (there is no pG4 on the plus strand in the shown region), SNS, ChIP H2A.Z and nucleosome coverage profiles, GC%, CpG islands and RefSeq RNA are shown. The NFR covering pG4#3 to 5 is delineated with a black line surrounded by red dotted lines (NFR). The med14 origin initiation site is indicated (ORI). **b** Endogenous

sequence found between pG4#1 and #5 upstream of the transcribed Med14 gene. Deletions previously made at the endogenous locus are indicated by red and blue rectangles for ΔpG4#5 and ΔpG4#4, respectively. **c** Sequences of the ectopic Med14 pG4#4 + 5 origin and Med14 pG4#4 + 5 minimal origin are shown on the top. The green rectangle indicates the region deleted to construct the minimal origin. Primers used for the quantification of SNS enrichments are indicated. SNS relative enrichments for both origins are shown ($n = 2$), mean values are indicated above each bar. Source data are provided as a Source Data file.

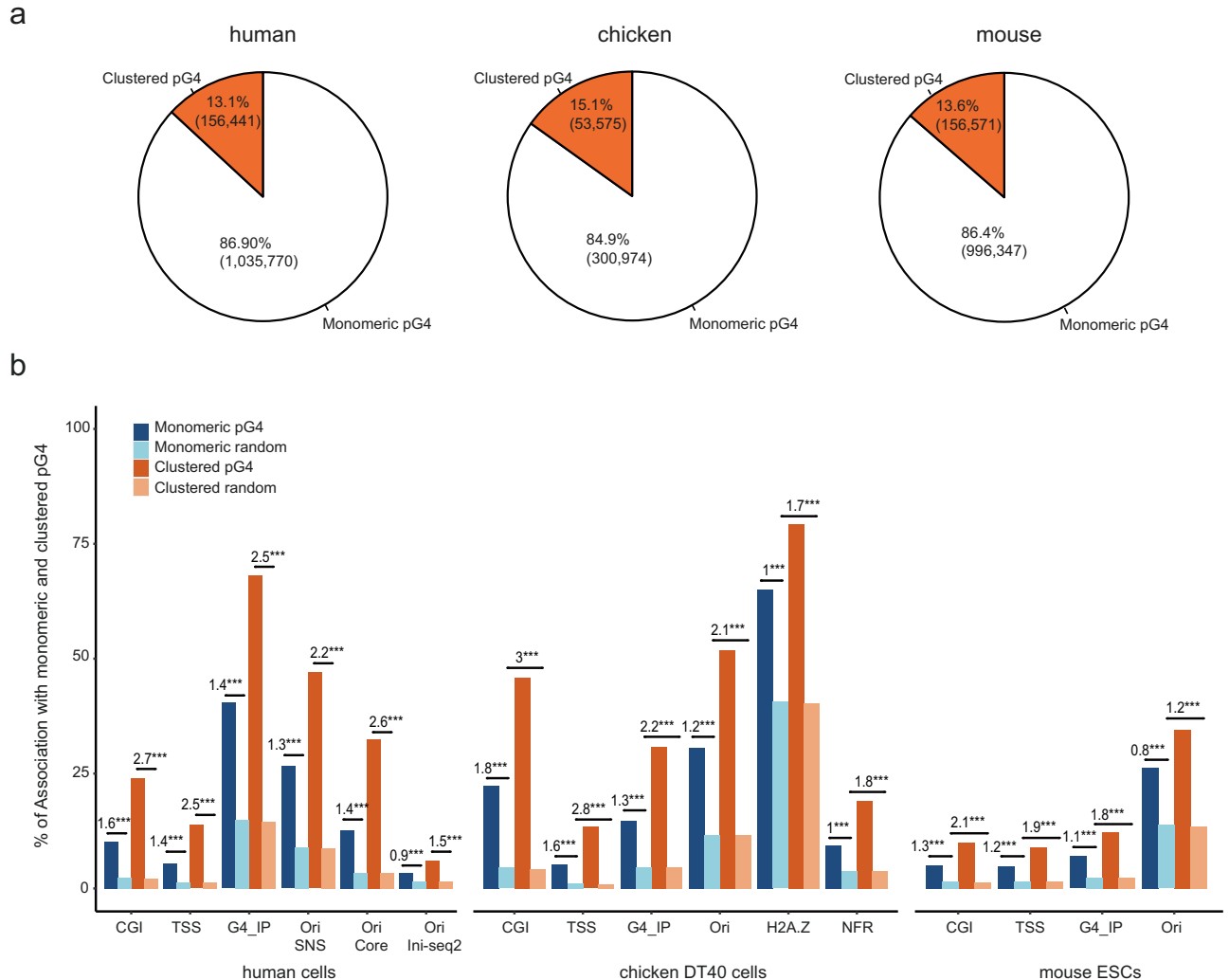

**Fig. 6 | Clustered pG4s have the same representation across vertebrates and are more associated with functional elements. a** Distribution of pG4s categories in the human (left), chicken (middle) and mouse (right) genomes. Clustered pG4 refer to ≥2 and ≤6 same-strand pG4s positioned <100 bp apart. **b** Associations of monomeric and clustered pG4s with specific genomic features in human, chicken and mouse cell lines. G4_IP data refers to G4 immunoprecipitation data collected using a small G4 probe that binds structured G4s in living cells with high efficiency and specificity[22]. Data obtained in four cell lines in human were pooled, whereas data in chicken were obtained in only one cell line. The numbers above bars indicate the log-odd ratio of the logistic regression to test enrichment with respect to random segments. ***$P < 0.001$ by the Wald test (see "Methods").

than monomeric pG4s (Fig. 6b). The result is confirmed in human cells with an antibody-based G4 chromatin immunoprecipitation dataset (Supplementary Table 1)[24]. This result suggests that there is a higher probability of finding a structured G4 in clustered pG4s than in isolated ones. Although an association between origins and pG4s has previously been established[8,11], further analysis of our data revealed a much greater enrichment of clustered pG4s compared with monomeric pG4s at origins, further supporting the putative functional role of clustered pG4s in replication initiation genome-wide (Fig. 7, Supplementary Tables 3 and 4, and Supplementary Materials). The associations between pG4s and origins do not depend on the detection method (SNS or Ini-seq2). Moreover, the strong association between clustered pG4s and Ini-seq2 detected origins shows that early S-phase origins are the most enriched in clustered pG4s. In agreement with the increased capacity of clustered pG4s to be folded, we observed that Ini-seq2 origins contain in their great majority a structured G4 (Fig. 7a). Finally, the 25% most active origins, responsible of ~80% of the SNS signal[23], are strongly associated with clustered pG4s, 38% and 36% in human and chicken, respectively. We also explored correlations between the number of monomeric and clustered pG4s at the origins and the strength of origin firing (SNS enrichment). While the strength

of origins (CGI and non-CGI) was positively associated with the abundance of both monomeric and clustered pG4s, the presence of clustered pG4s had a much stronger influence on origin activity (Fig. 7c). Indeed, the presence of up to two clustered pG4s at a non-CGI rendered its activity similar to that of the more active CGI origins (Supplementary Fig. 11B). Altogether, these results suggest an important role of clustered pG4s in the establishment of the replication program in vertebrates.

## NFR formation by clustered pG4s is associated with origin activity genome-wide

We next used MNase-Seq to investigate nucleosome coverage around pG4s across the chicken genome. We found that clustered pG4s associated or not with origins were significantly depleted of nucleosomes at the 5' side within CGIs. However, clustered pG4s associated with origins showed a stronger nucleosome coverage on the 3' side, generating asymmetric profiles (Fig. 8a, clustered pG4 CGI panel). Note that, as in our minimal origin, the NFR is located 5' of the G tract (Fig. 8a, b). The difference in nucleosome coverage at replication origins around clustered pG4s was even more pronounced outside of CGIs, as illustrated by the fact that non-origin-associated clustered

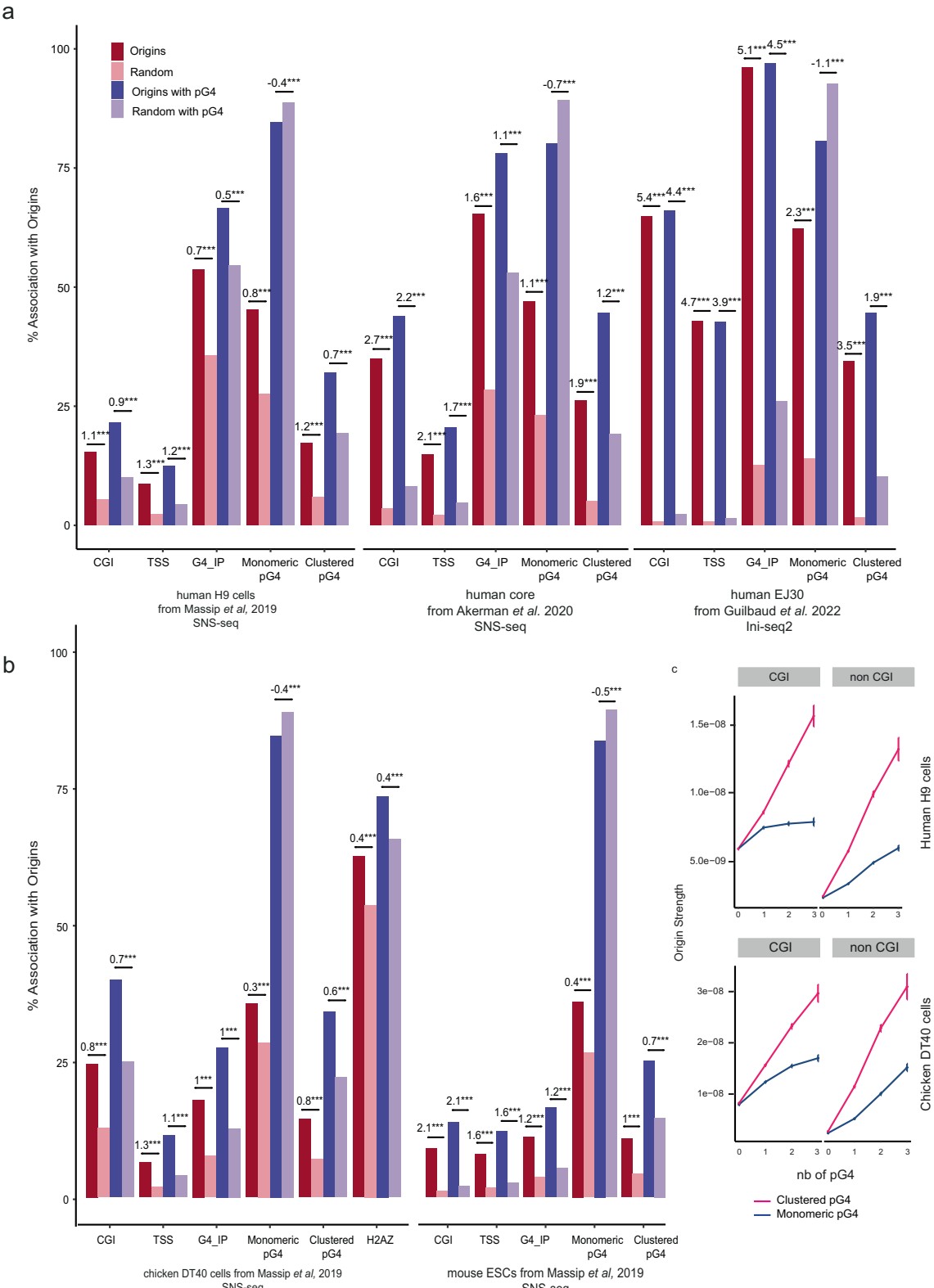

**Fig. 7 | Replication origins are more enriched in clustered pG4s than in monomeric pG4s. a**, **b** Association of origins peaks with genomic features. **a** SNS Origin peaks data come from either human H9 cells (left), the SNS core origins defined in ref. 23 (middle) or Ini-seq2 datas from ref. 16. **b** SNS Origin peaks data come from chicken DT40 cells (left) or mouse ESCs (right). The numbers above bars indicate the log-odd-ratio of the logistic regression to test enrichment with respect to random segments. ***$P < 0.001$ by the Wald test (see "Methods"). The negative LOR indicates that when origins are associated with a pG4s, the enrichment is towards the clustered form (LOR > 0) rather than the monomeric form (LOR < 0). **c** Origin strength detected in CGI (left) and non-CGI (right) is related to the number of monomeric (blue) and clustered (pink) pG4 associated in human and chicken cell lines. Origins strength is defined as the number of reads mapping within the origin divided by the length of the origin. Error bars correspond to a 95% two-sided confidence interval. The number of origins is provided in Supplementary Tables 3 and 4.

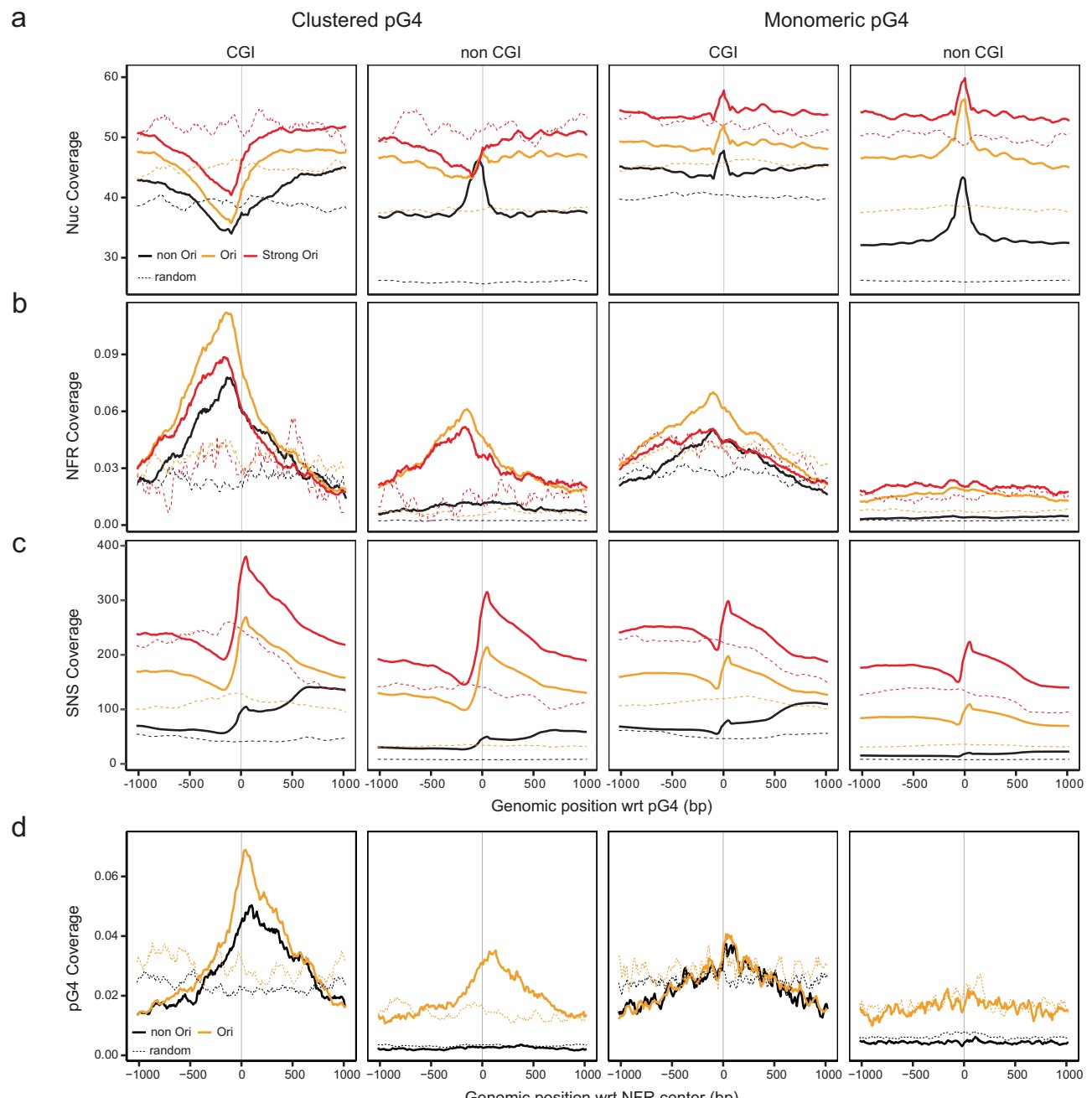

**Fig. 8 | Clustered pG4s are associated with NFRs located next to well-positioned nucleosomes genome-wide. a** Nucleosome coverage (number of reads in the input sample from MNase-Seq) around clustered and monomeric pG4s on the plus strand with respect to their association with origins (Ori/non-Ori). Strong origins are defined as the top 25% most active origins in terms of SNS enrichment. Random origins are as defined in "Methods". Vertical gray lines correspond to the first position in pG4s. **b** NFR coverage (number of peaks) around clustered and monomeric pG4s on the plus strand of cells in G1. **c** SNS coverage (number of reads) around clustered and monomeric pG4s on the plus strand. **d** Clustered and monomeric pG4 coverage around NFRs on the plus strand. Coverage plots represent the average number of genomic features in sliding windows of 20 bp, overlapping by 10 bp.

pG4s were enriched in nucleosomes whereas origin-associated clustered pG4s marked a transition between an NFR and well-positioned nucleosomes (Fig. 8a, clustered pG4 non-CGI panel). Moreover, all monomeric pG4s showed a clear enrichment in nucleosomes, irrespective of their proximity to origins, indicating that the depletion in nucleosomes is specific to clustered origin-associated pG4s (Fig. 8a, monomeric pG4 panel). Similar results were obtained when the complementary strand was analyzed, and these data were further confirmed by NFR coverage profiles obtained by ATAC-Seq (Fig. 8b and Supplementary Fig. 12A–C). Remarkably, the ATAC-seq data also produce asymmetric profiles only at clustered pG4s associated with origins at CGIs (Fig. 8b, clustered pG4 CGI panel). Consistent with the observations with our model origin (Fig. 4a), NFRs were already formed in G2, whereas pre-RCs were not assembled on chromatin (Supplementary Fig. 11A). Collectively, these results show that the presence of clustered pG4s correlates with the formation of an NFR next to well-positioned nucleosomes when associated with strong replication initiation activity (Fig. 8a–c).

An investigation of the genome-wide association of NFRs with functional elements revealed strong colocalization of NFRs with origins (~40% total, 16% with strong origins) and with pG4s (30% with monomeric pG4s, 12% with clustered pG4s) in chicken cells

(Supplementary Fig. 11B). The distribution of pG4s around NFRs was strongly dependent on the proximity to CGIs. Within CGIs, NFRs showed a significant peak of clustered pG4s regardless of the association with origins; in contrast, outside of CGIs, only origin-associated NFRs showed significant enrichment in clustered pG4s (Fig. 8d [clustered pG4 panel] and Supplementary Fig. 12D). Notably, NFRs showed no particular accumulation of monomeric pG4s as compared with random sequences. These data indicate that clustered pG4s are critical for the establishment of functional origins through NFR formation (Fig. 8d [monomeric pG4 panel] and Supplementary Fig. 12D).

### The histone variant H2A.Z is associated with pG4s genome-wide but does not predict origin activity

Consistent with the established regulatory role of H2A.Z in origin selection[13], we found that H2A.Z was broadly associated with replication origins ( ~ 60%) and with pG4s (65% with monomeric and 79% with clustered) in chicken cells (Figs. 6b, 7b, and 9a). In contrast to NFRs, H2A.Z did not exhibit a specific accumulation pattern around either monomeric or clustered pG4s (Fig. 9b). Conversely, the presence of clustered pG4s was not associated with H2A.Z accumulation around either origins or NFRs, which suggested that the presence of H2A.Z was insufficient to induce formation of an origin and NFR (Fig. 9c and Supplementary Fig. 12E). The lack of a specific H2A.Z pattern around clustered pG4s genome-wide is in agreement with our genetic analyses showing that H2A.Z is recruited at pG4s independently of the formation of an efficient origin or NFR (Fig. 4). The high proportion of H2A.Z peaks not associated with a replication origin (74% in human, 71% in chicken) points to a role for additional regulatory elements, such as the formation of an NFR and the presence of clustered pG4s, in defining an origin of replication (Fig. 9d). This is consistent with the strong enrichment of H2A.Z peaks at NFRs (73%, Supplementary Fig. 11B). Taken together, these genome-wide analyses suggest that H2A.Z recruitment is insufficient to define a functional origin.

## Discussion

In the present study, we sought to investigate the requirements for pG4s, NFRs, and H2A.Z in the formation of replication origins in chicken, human and mouse cells. Using genetic dissection of a model origin, the chicken β^A-globin promoter, we showed that the presence of two pG4s on the same strand within about 100 bp, which is the length of DNA wrapped around a single nucleosome, is sufficient to form an efficient origin. Analyses of chromatin structure showed a causal link between the formation of an NFR at the pG4 clustered and the formation of an active origin. Further, we demonstrated that an array of well-positioned nucleosomes carrying the histone variant H2A.Z, which is known to be a key regulatory factor for origin function, is present downstream of this NFR at the replication initiation site[13]. The med14 promoter/origin has a similar chromatin organization and two critical pG4s found on the same strand are also sufficient to form a minimal functional origin. We showed that this clustered pG4 cis-signal correlates with a significant fraction of replication origins as well as with the presence of an NFR next to well-positioned nucleosomes containing the H2A.Z variant. Thus, the results of this study establish a direct link between the presence of cis-elements and the formation of a significant fraction of origins of replication (including ~35% of strong origins), in vertebrates, and identify the crucial role of chromatin organization in this association. The recent observation that in yeast S. cerevisiae, a nucleosome next to an NFR is a preferential substrate for origin licensing led the authors to propose that this nucleosome-directed origin licensing paradigm generalizes to all eukaryotes[5]. Our results confirm this hypothesis although our model includes pG4s as another important player in vertebrates. The stronger association of clustered pG4s with structured G-quadruplexes suggests a role for structured G4s in origin activity in agreement with our previous study[9]. It remains to understand how they might act. Do structured G4s

promote the formation of NFRs or conversely, would the presence of an NFR promote the formation of structured G4s[24]? NFR formation is neither a necessary nor a sufficient signal to assemble an efficient origin since only 12% of origins are associated with an NFR and inversely 40% of NFRs are associated with an origin in chicken. The presence of the histone variant H2A.Z was also shown to be important[13]. Accordingly, we found that 62% of origins are associated with this histone variant in chicken cells. However, the large fraction of H2A.Z enriched peaks devoid of replication origins (~60%) reveals that this mark is not sufficient to drive efficient replication initiation. Overall, our results point to the involvement of a complex interplay between several actors in the definition of efficient origins. Key replication factors, such as the origin recognition complex and Mdm2 binding protein, are known to recognize structured G4s, which reinforces the important role of clustered pG4s in the formation and initiation of vertebrate replication origins[17,25,26]. Collectively, these results highlight the essential nature of cis-elements that organize the replication of vertebrate genomes through the establishment of a canonical chromatin structure, and additionally provide a foundation for further studies aimed at understanding the mode of action of clustered pG4s on the dynamics of vertebrate genomes.

## Methods

### Plasmid construction

The targeting vectors used for homologous recombination in DT40 cells were constructed with the multisite Gateway Pro kit (Thermo Fischer Scientific #12537100)[20]. Vectors containing the 5' and 3' target arms used for specific recombination at the mid-late genomic insertion site (chr1:72,565,520 bp, galGal5) were described previously[20]. We used four entry vectors to generate either the new β^A-globin +β-actin-BsR construct or the Med14 construct inserted at the mid-late genomic site: two entry vectors containing the 5' and 3' target arms for specific insertion, one entry vector (β-actin-BsR) containing the β-actin promoter linked with the blasticidin resistance gene (BsR), flanked by loxP sites and one entry vector pDONR221 P5-P4 containing either the β^A-globin fused to the IL2R gene and SV40 Poly-A sequence, flanked with two USF binding sites or the Med14 pG4#4 + 5 origin sequence. The corresponding final vector was generated by recombining compatible att sites between the entry vectors, with LR clonase (Thermo Fisher Scientific #12538120). For electroporation, the final vector was linearized with ScaI (NEB #R3122S). β^A-globin and Med14 origins mutagenesis were made using overlapping primers replicating the entire entry vector, using the Herculase II fusion DNA polymerase (Agilent #600679) according to the manufacturer recommendations. Primers were designed with around 15 bp of overlapping sequence to ensure proper circularization with the In-Fusion HD cloning plus kit (Takara #638909). Mach1 competent cells (Thermo Fisher Scientific #C862003) were used for plasmid cloning.

### Cell culture condition and transfection

DT40 cells were grown in RPMI 1640 medium supplemented with Glutamax (Thermo Fisher Scientific #61870010), containing 10% FBS, 1% chicken serum, 0.1 mM β-mercaptoethanol, 200 U/mL penicillin, 200 μg/mL streptomycin and 1.75 μg/mL of amphotericin B at 37 °C, under an atmosphere containing 5% $CO_2$. Cells were electroporated as previously described[20]. Cell clones were selected on media containing a final concentration of 20 μg/ml blasticidin. Genomic DNA was extracted from cells in lysis buffer (10 mM Tris pH 8.0; 25 mM NaCl; 1 mM EDTA and 200 μg/mL proteinase K). Clones into which the plasmid DNA was integrated were screened by PCR with primer pairs designed to bind on one side of the insertion site such that one primer bound within the construct and the other primer bound just upstream or downstream from the arm used for recombination (Supplementary Fig. 14 and Supplementary Table 9).

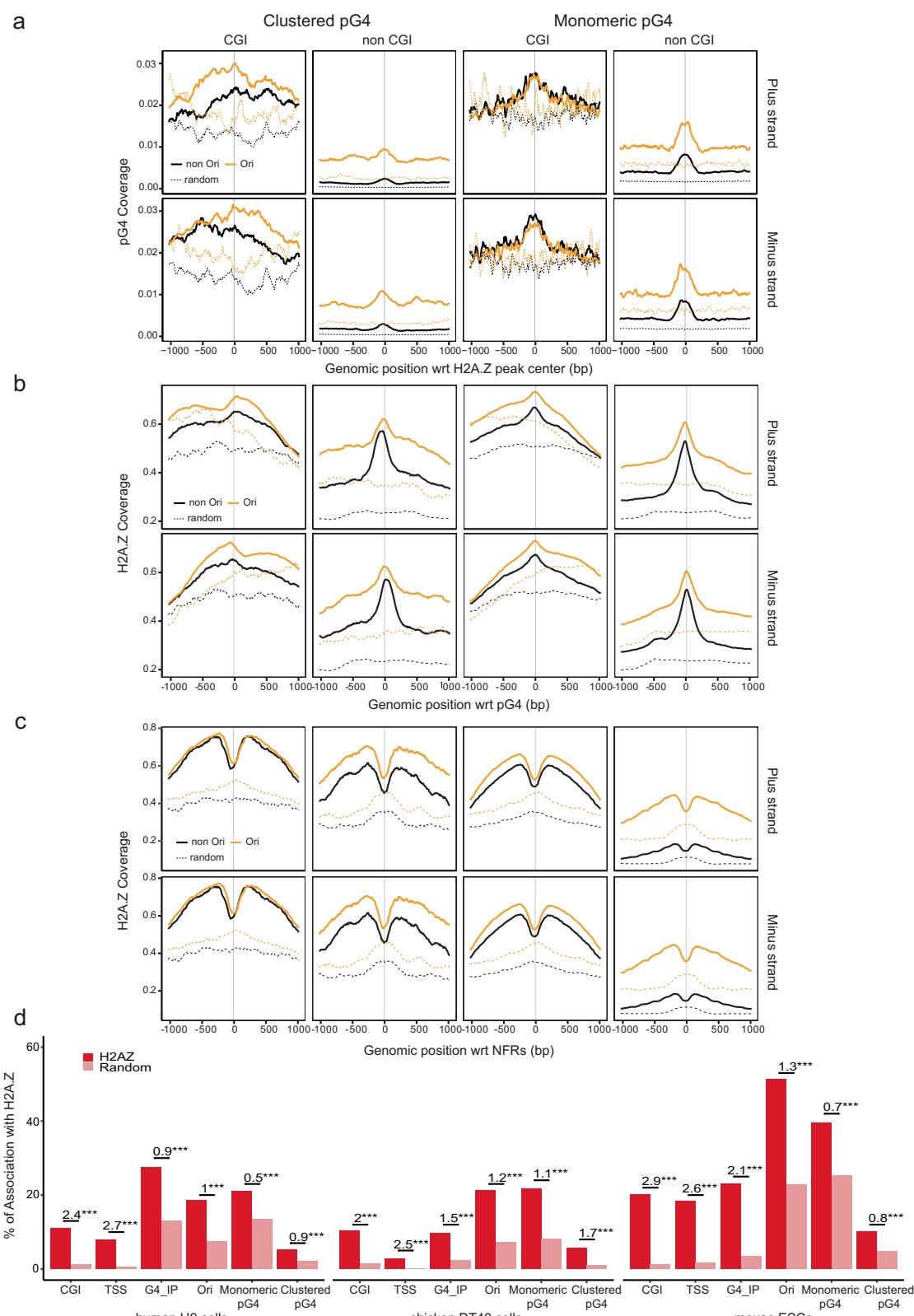

**Fig. 9 | H2A.Z is significantly associated with replication origins and pG4s genome-wide but not sufficient to induce the formation of an efficient origin.** **a** Clustered and monomeric pG4 coverage around H2A.Z peaks on the plus (top) and minus (bottom) strands. **b** H2A.Z coverage (number of reads) around clustered and monomeric pG4s on the plus (top) and minus (bottom) strands. **c** H2A.Z coverage (number of reads) around NFRs containing clustered and monomeric pG4s on the plus (top) and minus (bottom) strands. **d** Associations of H2A.Z peaks with genomic features in human, chicken and mouse cells. Numbers above bars indicate the log-odd ratio of the logistic regression to test enrichment with respect to random segments. ***$P < 0.001$ by the Wald test (see "Methods"). Coverage plots represent the average number of genomic features in sliding windows of 20 bp, overlapping by 10 bp. See Fig. 8 for additional definitions.

The *BsR* resistance gene was excised from positive clones using the Cre-LoxP system. DT40 cells constitutively express a tightly regulated Cre recombinase fused to a mutated estrogen receptor (Mer)[27]. This inactive Mer-Cre-Mer fusion protein can be transiently activated in the presence of 4-hydroxytamoxifen, resulting in the efficient excision of genomic regions flanked by two recombination signals (*loxP* sites) inserted in the same direction. For the excision of the genomic DNA flanked by loxP sites, we treated $3 \times 10^5$ cells with 5 μM 4-hydroxytamoxifen (Sigma-Aldrich #T176) for 24 h. Subclones were obtained by plating dilutions of the treated cell suspension at a density of 50, 150, and 1500 viable cells per 10 ml in 96-well flat-bottomed microtiter plates. Genomic DNA was extracted from single subclones and analyzed by PCR with specific primer pairs (Supplementary Table 9). We assessed the excision of the β-actin-BsR selection cassette with a primer pair amplifying the poly-A sequence of the both *BsR* and *Il2R* genes and the downstream part of the 3' arm of the insertion site (Supplementary Fig. 14C, F, primer pair #3/#4, Supplementary Table 9). All clones were cultured for 72 h in selective media containing the appropriate antibiotic, to confirm PCR results. For each clonal line, the copy number of the construct was quantified by qPCR (Supplementary table 10). Homozygous insertions were made in a two-step process. Cells were first heterozygously modified as described earlier with excision of the BsR gene. Once cell lines were established, a second homologous recombination was made to target the other chromosome. Cells were then double screened, one to confirm the maintenance of the first inserted construct (Supplementary Fig. 14E, primer pair #5a/6) and the second to validate proper insertion of the second construct (Supplementary Fig. 14E, primer pair #5b/6).

## SNS purification
SNS purification was performed as previously described[9] with some slight modifications. Fresh cultured cells were used for total genomic DNA extraction, and the T4 polynucleotide kinase (Biolabs #M0201S) concentration was adjusted to 100 U and incubated for 30 min at 37 °C. Proteinase K (Thermo Fischer Scientific #EO0491) digestion was realized at a final concentration of 625 μg/ml for 30 min at 50 °C.

## Replication timing analysis
For RT experiments, about $10^7$ exponentially growing cells were pulse-labeled with 5-Bromo-2'-deoxyuridine (BrdU, Sigma-Aldrich #B9285) for 1 h and sorted into four S-phase fractions, from early to late S phase. The collected cells were treated with lysis buffer (50 mM Tris pH 8.0; 10 mM EDTA pH 8.0; 300 mM NaCl; 0.5% SDS, 0.2 mg/ml of freshly added proteinase and 0.5 mg/ml of freshly added RNase A), incubated at 56 °C for 2 h and stored at −20 °C, in the dark. Genomic DNA was isolated from each sample by phenol−chloroform extraction and alcohol precipitation and sonicated four times for 30 s each, at 30 s intervals, in the high mode at 4 °C in a Bioruptor water bath sonicator (Diagenode), to obtain fragments of 500−1000 bp in size. The sonicated DNA was denatured by incubation at 95 °C for 5 min. We added monoclonal anti-BrdU antibody (BD Biosciences #347580) at a final concentration of 3.6 μg/ml in 1× IP buffer (10 mM Tris pH 8.0, 1 mM EDTA pH 8.0, 150 mM NaCl, 0.5% Triton X-100, and 7 mM NaOH). We used 50 μl of protein-G-coated magnetic beads (from Thermo Fisher Scientific #10004D) per sample to pull down the anti-BrdU antibody. Beads and BrdU-labeled nascent DNA were incubated for 2−3 h at 4 °C, on a rotating wheel. The beads were then washed once with 1x IP buffer, twice with wash buffer (20 mM Tris pH 8.0, 2 mM EDTA pH 8.0, 250 mM NaCl, 0.25% Triton X-100) and then twice with 1× TE buffer pH 8.0. The DNA was eluted by incubating the beads at 37 °C for 2 h in 250 μl 1× TE buffer pH 8.0, to which we added 1% SDS and 0.5 mg/ml proteinase K. DNA was purified by phenol−chloroform extraction and alcohol precipitation and resuspended in 50 μl TE. For Repli-seq analyses, immunoprecipitated NS from the four S-phase fractions collected by flow cytometry or from an asynchronous cell population

were amplified by whole-genome amplification (GenomePlex Complete Whole Genome Amplification kit #WGA2; Sigma) according to the manufacturer's recommendations to obtain sufficient DNA amount. After amplification, libraries were constructed as described in the sequencing library preparation section. The BrdU-Labeled nascent strands (NS) quantification was performed, as previously described[9,18,20]. RT shifts were calculated as NS enrichment changes occurring on late fractions ($\Delta L = [\%(S3 + S4)_{with} - [\%(S3 + S4)_{without}]$) and on the earliest fraction ($\Delta E = [\%S1_{with} - \%S1_{without}]$) following this formula: RT shift = - $\Delta L + \Delta E$. The higher the RT shift is and the more the replication time advanced (Supplementary Fig. 2).

## Repli-Seq data processing
Paired-end sequencing data were mapped on the galGal5 chicken genome using bowtie2 version 2.3.4.1. For each timing fraction, replication timing was computed using 50 kb sliding windows at 10 kb intervals, normalized by the global and local genomic coverage of the asynchronous cell population, to normalize for total and local coverage variations. Then the centered and standardized timing profiles were smoothed using cubic splines (smooth.spline function of R). In order to provide a single RT profile combining all fractions, we used the method proposed in ref. 28, by computing the weighted average WA = (0.750*S1) + (0.583*S2) + (0.417*S3) + (0.250*S4). An increase in WA indicates an earlier timing.

## Flow cytometry analysis
After BrdU incorporation, DT40 cells were washed twice with PBS, fixed in 75% ethanol, and stored at −20 °C. On the day of sorting, fixed cells were resuspended at a final concentration of $2.5 \times 10^6$ cells/mL in 0.1% IGEPAL in PBS (Sigma, #CA-630), 50 μg/ml propidium iodide and 0.5 mg/ml RNase A, and incubated for 30 min at room temperature. Singlet cells were sorted with an INFLUX 500 cell sorter (Cytopeia, BD Biosciences) or a FACSAria Fusion (BD Biosciences) (Supplementary Fig. 16). Four fractions of S-phase cells (S1−S4), each containing $5 \times 10^4$ cells, were collected and further treated for locus-specific RT analyses.

## MNase digestion
We cross-linked $30 \times 10^6$ exponential growing cells by incubation for 5 min with 1% freshly prepared formaldehyde (Thermo Fisher Scientific, #28908) at room temperature. Fixation was stopped by adding 0.125 M glycine-PBS for five minutes at room temperature. After three ice-cold PBS washing, nuclei were extracted in lysis buffer (10 mM Tris-HCl, pH 7.5, 10 mM NaCl, 3 mM MgCl₂, 0.2% Triton X-100, 0.5 mM EGTA, 1 mM DTT, 1× protease inhibitor cocktail (Sigma, #P8340)) for 5 min on ice, centrifuged and resuspended in digestion buffer (10 mM Tris-HCl pH 7.5, 10 mM NaCl, 3 mM MgCl₂, 1 mM CaCl₂, 1× protease inhibitor cocktail (Sigma, #P8340)). Micrococcal Nuclease (MNase; Thermo Fisher Scientific, #EN0181) digestions were performed for 15 min at 37 °C, using a series of four increasing concentrations of MNase (2.5, 10, 40, and 160 U/mL). The lowest MNase concentration generated a mixture of oligo-, di- and mono-nucleosomes, whereas the highest concentration produced mostly mono-nucleosomes (supplementary Fig. 15). The final concentration of 160 U/mL was used for ChIP and nucleosome coverage analyses. The reaction was stopped by adding 0.1 volume of stop buffer (200 mM EDTA pH 8.0, 40 mM EGTA pH 8.0). Chromatin was then sonicated during 30 s at 7 °C in a buffer adjusted to a final SDS concentration of 0.1% with a Covaris sonicator using 75 W intensity, 200 cycles per burst, 5% of duty cycle. For nucleosome coverage analyses, DNA molecules were recovered after RNAse A and proteinase K digestion and phenol−chloroform extraction and ethanol precipitation. For ChIP analyses, chromatin was adjusted in 1× IP buffer (20 mM Tris-HCl pH 8.0, 2 mM EDTA pH 8.0, 150 mM NaCl, 1% Triton X-100 and 0.1% SDS).

## Chromatin immunoprecipitation (ChIP)

Immunoprecipitation was performed overnight at 4 °C, in a final volume of 300 μl of 1× IP buffer (20 mM Tris-HCl pH 8.0, 2 mM EDTA pH 8.0, 150 mM NaCl, 1% Triton X-100 and 0.1% SDS) on an amount of MNAse digested chromatin corresponding to 10 μg of DNA, with anti-H2A.Z antibody (Active Motif, #39013), according to manufacturer recommendations. Immuno-complexes were pulled down with 50 μl of protein-G coated magnetic beads (Thermo Fisher Scientific, Dynabeads protein G, #10004D) per sample. Beads and immuno-complexes were incubated for 2 h at 4 °C, on a rotating wheel. The beads were then washed once with 1× IP buffer, twice with wash B buffer (20 mM Tris pH 8.0, 2 mM EDTA pH 8.0, 500 mM NaCl, 0.25% Triton X-100) and then twice with 1× TE buffer pH 8.0. The DNA was eluted by incubating the beads for 2 h at 37 °C with 250 μl 1× TE buffer pH 8.0, to which we added 1% SDS and 0.5 mg/ml proteinase K. Cross-linking was reversed by overnight incubation at 65 °C, and samples were further treated with 10 μg of RNase A for 15 min at 37 °C, and with 20 μg of proteinase K for 1 h at 56 °C. DNA was purified by phenol–chloroform extraction, precipitated in alcohol, resuspended in 100 μl TE and quantified by QuBit (dsDNA HS assay kit, Thermo Fisher Scientific, #Q32851). Input samples were obtained from MNAse digested chromatin after RNASe A and proteinase K digestion, phenol–chloroform extraction and ethanol precipitation. These samples were used as a reference for H2AZ peak detection, performed using bowtie2 version 2.3.4.1 and MAC2s version 2.1.2.

## RNA extraction and reverse transcription

Total RNA were extracted from $5 \times 10^6$ cells with the Nucleospin RNA kit (Macherey Nagel, #740955). 20 μg of total RNA was then treated with four units of DNAse I (NEB, #M0303S) for 1 h at 37 °C. The enzyme was inactivated by adding 5 mM EDTA and incubating the reaction mixture for 10 min at 75 °C. The RNA was then purified by phenol–chloroform extraction and ethanol precipitation. Reverse transcription reactions (RT +) were then performed with 5 μg of RNA and random hexamers (NEB, #S1330S), using the Superscript III Reverse Transcriptase (Thermo Fisher Scientific, #18080093) according to the manufacturer's instructions. Negative controls (RT−) were performed with the same procedure, but without the addition of reverse transcriptase. The comparison of RT+ and RT− samples was used to validate DNAse I treatment and the complete digestion of the genomic DNA in the RNA samples.

## Real-time PCR quantification of DNA

Real-time qPCRs were executed according to the MIQE guideline. The Techne Prime Pro48 apparatus and the QPCR-SYBR Green mix (Thermo Fisher Scientific, #AB1285B) were used for the real-time PCR quantification of BrdU-labeled nascent strands (NS), genomic DNA extracted from 4-hydroxytamoxifen-treated clonal cell lines, short nascent strands or cDNA. Each sample was quantified at least in duplicates. For all reactions on the Techne Prime Pro48, real-time PCR was performed under the following cycling conditions: initial denaturation at 95 °C for 15 min, followed by 50 cycles of 95 °C for 15 s, 61 °C for 30 s, 72 °C for 20 s, and fluorescence measurement. Following PCR, a thermal melting profile was used for specific amplicon validation.

## Cells sorting by centrifugal elutriation

Elutriation was performed as previously described in[29] with slight modifications. We used a Beckman Coulter Avanti® J-26 XP with a JE-5.0 rotor. For MNAse experiments, around $1.5 \times 10^9$ cells were resuspended into 200 ml of elutriation buffer (PBS1×, 1% FBS and 1 mM EDTA; filtered on 0.22 μm). Cells are injected into a large elutriation chamber of 40 ml subjected to a rotation of 2700 RPM, with a pump flow of 80 ml/min. Once injected, cell population was incubated for 30 min into the chamber for recovery and equilibration. Cells are sorted by size with decreasing centrifugal speed. We recover five fractions of cells (from F1 to F5) with the following parameters of centrifugal force/recovered volume: 2530 RPM/400 ml; 2380 RPM/800 ml; 2260 RPM/800 ml; 2130 RPM/800 ml; 1980 RPM/800 ml. Fraction 2 corresponds to G1 cells, and the fraction 5 to G2/M cells. Cells were then cultured with classical medium culture at a concentration of $1 \times 10^6$ cells/ml. They were either directly treated with L-mimosine for 3 h, or directly cross-linked and extracted for MNase digestion. Cells synchronization in G1/S was performed using 160 μg/ml of L-mimosine for 3 h (Sigma-Aldrich #M0253).

For ATAC experiments, around $3 \times 10^8$ cells were resuspended into 100 ml of elutriation buffer and injected into a standard elutriation chamber of 4 ml subjected to a rotation of 3800 RPM, with a pump flow of 40 ml/min. We recover six fractions of cells (from F1 to F6) with the following parameters of centrifugal force/recovered volume: 3600 RPM/150 ml; 3400 RPM/150 ml; 3200 RPM/150 ml; 3000 RPM/150 ml; 2800 RPM/150 ml; 2600 RPM/150 ml. Fraction 2 corresponds to G1 cells, and the fraction F6 to G2/M cells. Cells were used directly for ATAC experiments.

## Assay for transposase accessible chromatin (ATAC) experiment

ATAC experiments were performed as described in ref. 30. Briefly, $1 \times 10^5$ asynchronous, G1 and G2/M cells collected by elutriation were washed once with cold PBS. The pellet was resuspended in 50 μl of Lysis buffer (0.3 M Sucrose, 10 mM Tris pH 8, 10 mM NaCl, 3 mM $MgCl_2$, 0.2% Triton X-100) and the transposition reaction was incubated for 30 min at 37 °C in a thermomixer shaking at 700 rpm. Samples were purified with the MinElute PCR purification kit (Qiagen #28004) following manufacturer instructions and eluted in 11 μl. The tagmentation and library preparation was performed with the Nextera DNA Library Preparation Kit with eight cycles of PCR amplification (Illumina #FC-121-1030).

## Sequencing library preparation

Sequencing libraries were prepared using NEB-Next Ultra II DNA library prep Kit for Illumina (NEB #E7645S) according to the manufacturer's instructions for MNase, ChIp and input libraries. Samples were not subjected to size selection, but only cleaned up for adaptor-ligated DNA using SPRISelect Reagent kit (Beckman coulter #B23317), to ensure a MNase digested small fragment overall recovery. Libraries were prepared with a starting DNA input of 100 ng for MNAse libraries and 10 ng for ChIp libraries. For ChIp libraries prep, we used a tenfold diluted adaptor for the adaptor ligation. The library amplification was performed using NEB-Next Multiplex Oligos for Illumina (NEB #E7710S) with different NEB-Next index primers and the NEB-Next Universal PCR Primer (NEB #E6861A) with three PCR cycles for MNAse samples or eight PCR cycles for ChIp samples. Sequencing libraries for ATAC samples were prepared using the Nextera DNA library preparation kit (Illumina, #FC-121-1030) according to the manufacturer's instructions. The tagmented DNA was amplified with eight cycles of PCR amplification and the libraries were cleaned up with SPRselect beads (Beckman Coulter, #B23317). The mean size of the library molecules and the quality of the libraries were determined on an Agilent Bio-analyser High Sensitivity DNA chip (Agilent Technologies, #5067–4626).

## Sequencing

Sequencing was made at the GENOM'IC Cochin institute facility (Paris) on a NextSeq 500 Illumina sequencer. Samples were sequenced with a High Output 150 cycles Flow Cell in paired-end (paired-end reads of 75 bp) according to standard procedures. For MNAse accessibility and H2A.Z ChIp analyses, 200 M of reads were generated. For ATAC analysis, 200 M reads were generated.

## Databases

CGI annotations were downloaded from the UCSC database[31] and TSSs from the Eukaryotic Promoter Database[32] for human and

mouse, and from the Ensembl website[33] for chicken. pG4 detections were downloaded from[22] and correspond to automatic detections combining pattern matching and manual curation. Clustered pG4s were detected using mergeBed of the BedTools toolset[34], with a maximum distance between pG4s allowed for pG4s to be merged of 100 bp (-d option), and from 2 to 6 instances of pG4s within these 100 bp. In addition to predicted pG4s, we also considered structured G4s detected by ChIP-Seq with a G4P protein (ref. 22, GSE133379) for human (293T, A549, H1975, HeLa-S3 cells), chicken (DF-1 cells) and mouse (3T3 cells). Human ChIP-G4s were combined. Finally, we also considered structured G4s identified by ChIP-seq with antibodies in human cells (combined peaks from HaCat and NHEK cells, GSE99205). Replication origins detected by SNS were downloaded from ref. 35, which correspond to human (H9), chicken (DT40) and mouse (mESC) data. Complementary analyses were performed on core origins from ref. 23 (GSE128477), also detected by SNS, and on origins detected by Ini-seq2 from ref. 16 (GSM5658908). Human H2AZ peaks were downloaded from ref. 13, and mouse H2AZ peak from ref. 36 (GSM984544).

### Statistical analysis

Overlaps between genomic features were performed using intersectBed of the BedTools[34], with genomic segments (pG4s, origins, H2AZ, NFR), 1 kb extended from their peak coordinate. Regarding the association of origins with pG4s, we inspected separately the association with pG4s ([+] strand) and pC4s ([−] strand) upstream and downstream origins peak, respectively. Random segments were sampled using the random function of BedTools[34] and match the size and the GC-content distribution of the corresponding feature. The significance of enrichments was assessed using logistic regression[37]. For instance, the analyses of replication origins associations rely on their relative enrichment in genomic features as compared with random segments. Logistic regression produces log-odds-ratios (logOR) of association: logOR=0 means that the association of this feature is the same in origins and random segments, if logOR>0 the association is stronger in origins, if logOR<0 the association is weaker in origins. Similar analyses were performed to assess the associations of pG4s in their mono or clustered form, H2A.Z and NFR with genomic features. In the case of origins, H2A.Z and NFR, we also computed their associations with other genomic features when they carry a pG4, as compared with random segments that also carry a pG4 (Supplementary Tables 3–6). These analyses show that when associated with pG4s, the relative enrichment of origins (or H2A.Z, or NFR) concerns clustered pG4s and not monomeric pG4s. Details on the analysis methodology are provided in the Supplementary Material. Coverage plots represent the average number of genomic features in sliding windows of 20 bp, overlapping by 10 bp.

### ATAC-seq

ATAC-seq reads were trimmed and then mapped to the chicken gal5 genome using version 2.3.4.1 of bowtie2. We removed discordant pairs (inserts >2000 bp) as well as reads with quality score below 40 and reads mapping to mitochondrial DNA. Finally, duplicated reads were removed using Picard Toolkit (http://broadinstitute.github.io/picard/, version 2.20). Reads from the three different replicates were pooled together to increase the statistical power. To do so, we first called open regions using macs2 (with option --broad), and peaks close to each other (<30 bp) were merged together. We then called NFRs in open regions using the nucleoatac pipeline[38].

### Reporting summary

Further information on research design is available in the Nature Portfolio Reporting Summary linked to this article.

## Data availability

All raw sequencing files and processed files generated in this study have been deposited in NCBI's Gene Expression Omnibus and are accessible through GEO Series accession number GSE231492. Additional data supporting the findings of this study are provided in the Supplementary Information files and the Source Data file. Source data are provided with this paper.

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

## Acknowledgements

The authors thank the members of the laboratory of M.-N.P. for useful insights and discussions. We thank S. Duharcourt, L. Duret, P. Gilardi-Hebenstreit, G. Felsenfeld, A. Piazza, and V. Studitsky for critical reading of the manuscript. We thank the ImagoSeine core facility of the Institut Jacques Monod platform, notably Magali Fradet and Nicolas Valentin for performing cell sorting. Sequencing of replication timing, MNAse-seq, and ATAC-seq data were performed by the GENOM'IC core facility at Institut Cochin-Paris. This work was supported by grants from the Association pour la Recherche sur le Cancer (ARC-Equipe Labellisée), and La ligue contre le cancer, comité d'Ile de France (2020–2022) obtained by M.-N.P. and the Agence Nationale pour la Recherche (ANR-15-CE12-0004-01) obtained by M-N.P and F.P. J.P.-B. was a recipient of a ligue contre le cancer doctoral fellowship. N.B. was a recipient of an ARC post-doctoral fellowship and N.P. of a post-doctoral fellowship from the LABEX Who Am I? M.-N.P. is supported by Inserm.

## Author contributions

J.P.-B. conducted most experiments. C.T.-D. performed MNase-seq and ChIP-seq experiments. A.-L.V. initiated the project, made and analyzed several mutants. M.L. analyzed transcriptional activity and SNS enrichment of several clones. M.G. analyzed nucleosome positioning along minimal origins. N.B. performed cell synchronization. N.P. made ATAC-seq experiments. F.M. and F.P. performed the statistical analyses. J.P.-B., C.T.-D., F.P., and M.-N.P. designed the experiments. J.P.-B., C.T.-D., F.P., and M.-N.P. wrote the paper with input from A.-L.V. F.P. and M.-N.P. supervised the project. F.P. and M.-N.P. obtained funding.

## Competing interests

The authors declare no competing interests.
