## [Peer Review File · Nature Communications]

Dimeric G-quadruplex motifs-induced NFRs determine strong replication origins in vertebratesEditorial Note: This manuscript has been previously reviewed at another journal that is not operating a transparent peer review scheme. This document only contains reviewer comments and rebuttal letters for versions considered at *Nature Communications*.

REVIEWER COMMENTS

Reviewer #1 (Remarks to the Author):

The authors are now providing more material including deeper genome wide analyses by confronting their data with other studies that straighten their conclusions. I still believe that widening the thoughts by analysing other non-B DNA structures would have been of great interest, but this latter comment shouldn't prevent this study to be published as of interest for the field.

I would however have few comments and questions on new figures 6 and 7, especially regarding Figure7 to clarify the findings for readership:

Figure6B:

-The legend "human H9 cells" is misleading as I believe it only refers to Ori SNS and not Ori core, Ori Iniseq(2) etc

-The authors should clearly detail in the figure legend what they refer to as G4_IP and link it with the used publication.

Figure7:

In the core manuscript, I understood that the authors confronted their data with data from Massip et al., Akerman et al. and Guilbaud et al. However, I am puzzled by the numbers of origins reported in the corresponding Supplementary Table3 that do not match the numbers in those publications:

- In, Massip et al., 155 395 origins are reported, where Table3 state 310,790.

- Number of all Core human origins from Akerman et al.: 64,000 core origins are reported and 127,248 in Table3.

- Iniseq2 origins from Guilbaud et al.: 23,905 reported origins Vs 47,634 in table3.

Thus, is this a mistake, misunderstanding? Or have the data been re-analysed with lowered thresholds, and how? I could not find the referred supplementary materials "p12: "(Fig. 7, supplementary tables 3 and 4, and supplementary materials)". But, if the data have been re-analysed, this should be clearly stated in the core manuscript and the figure itself, as this is not quite the same as only using published materials. Readers need to appreciate if those analyses have been carried in agreement/or not, and if so how, with the dataset/stringencies reported by the initial publications.

Minor points:

The authors should use "Iniseq2" wherever referring to Guilbaud et al. as I understood that Iniseq refers to Langley et al.

Reviewer #2 (Remarks to the Author):

In this revised manuscript by the Prioleau's group, they have addressed in a satisfactory and appropriate manner the issues pointed by me and other referees. The findings that the presence of

the two tandem G4-forming sequences (or clustered pG4s) is an important cis-element for active origins and that it is strongly associated with adjacent NFR add important novel information regarding the constituents required for vertebrate origin functions, although the findings point to many unanswered questions including what kind of higher-order structure are generated by the tandem pG4s at the origins, how they are generated and how they would facilitate origin firing, and how they are associated with efficient formation of NRF.

Other comments:

1. The authors state in the abstract that "Dimeric pG4s in replication origins trigger formation of an NFR next to precisely-positioned nucleosomes enriched in H2A.Z on this minimal origin and genome-wide."

However, they also state in Discussion that "It remains to understand how they might act. Do G4s promote the formation of NFRs or conversely would the presence of an NFR promote the formation of G4s?" Therefore, it is probably not accurate to state that "Dimeric pG4s in replication origins trigger formation of an NFR--". It would be more accurate to state at this stage that "Dimeric pG4s in replication origins are associated with formation of an NFR—".

2. Figure 3D is helpful. However, the nomenclatures of constructs are still complicated. If authors can add schematic drawing of the constructs in the table (just to indicate the positions and orientations of pG4 and other features as well as the locations of the mutations introduced), it would be very helpful to the readers to visually grasp the data and conclusions.

Reviewer #3 (Remarks to the Author):

In this paper, replication initiation activity was measured in several constructs containing guanine-rich motifs inserted into the beta-A globin locus in chicken DT40 cells and in another potential origin region at the Med14 promoter. The paper builds upon a previous publication (2014), which demonstrated a correlation between the formation of stable G-quadruplexes and replication origin activity. Additional constructs are reported, and it is shown that several combinations of G-quadruplex forming sequences at various intervals, and in particular, constructs that included two G-quadruplex forming sequences on the same strand, also facilitated replication origin activity. In line with previous studies, the insertion of these constructs was also associated with the formation of nucleosome free regions and some enrichment of histone H2AZ adjacent to replication origins. The paper also includes an analysis of genome-wide abundance of G-quadruplex forming sequences, utilizing primarily published datasets, demonstrating that clustered G-quadruplex forming sequences are enriched at replication origins and in the vicinity of "functional" elements, although they can be also present at non-origin regions.

This is a revised submission. The addition of several data points and some clarifications in view of the observations reported in the previous studies were useful. However, some of the concerns raised by the reviewers of the previous version remain unaddressed. Specific comments are listed below:

1. Abstract. As currently written, the abstract does not explicitly state that the role of dimeric G-quadruplex forming sequences was investigated in a model system at select loci in chicken cells. Because it is becoming increasingly evident that replication origins are a diverse group and that replication initiation is oftentimes an indicator of chromatin accessibility, the association between specific chromatin features and replication initiation can be pertinent only to some select loci. The claim that the data "suggest a crucial role for dimeric pG4s in the organization and duplication of vertebrate genomes" requires further substantiation.

2. H2A.Z data. In agreement with the opinion of the reviewer #2, the conclusion that H2A.Z recruitment is controlled by pG4 density requires further substantiation.

3. Strong origins. The definition of strong origins at the top 25% of identified origins in terms of SNS

enrichment is acceptable, however, it is unclear how G-quadruplex forming sequences can determine strong replication origins if they only associate with 16% of those origins. How are other origins determined? Do G-quadruplex forming sequences play another role? These issues should at least be discussed.

4. Origin efficiency. Related to the above, SNS enrichment of 30% is interpreted as strong origin activity in Figure 3 but as weak in Figure 1. This should be clarified.

5. Structured G-quadruplexes. It would be good to define this term unambiguously and use consistently throughout the paper.

6. Figure 1b. I was only able to see SNS tracks on the left-hand panel. It is crucial to show the SNS patterns for this construct, which forms the basis of the subsequent analyses.

7. Figure 5a shows a very strong enrichment of histone H2A.Z in a region that is identified as a nucleosome free region. How is this observation explained? Is there sufficient resolution to delineate the boundaries of the nucleosome free regions?

8. Related to the above, the data suggest that nucleosome free regions are shown in both origin-associated and non-origin areas. How does this observation relate to the notion that NFRs determine origin activity? This observation should at least be discussed.

9. Figure 9. As also pointed out by another reviewer in the previous review, the association between H2A.Z and origins is very modest and does not strongly support the figure title.

Minor.

Figure 1c is cited in the text before figure 1b.

Figure 5A, the text is too small and it would be good to include more details in the legend.

REVIEWER COMMENTS

Reviewer #1 (Remarks to the Author):

The authors are now providing more material including deeper genome wide analyses by confronting their data with other studies that straighten their conclusions. I still believe that widening the thoughts by analysing other non-B DNA structures would have been of great interest, but this latter comment shouldn't prevent this study to be published as of interest for the field.

I would however have few comments and questions on new figures 6 and 7, especially regarding Figure7 to clarify the findings for readership:

Figure6B:

-The legend "human H9 cells" is misleading as I believe it only refers to Ori SNS and not Ori core, Ori Iniseq(2) etc

We thank the referee for his detailed reading. We have modified the legend and put only human cells.

-The authors should clearly detail in the figure legend what they refer to as G4_IP and link it with the used publication.

We added in the legend a detailed explanation of what G4_IP refers to.

Figure7:

In the core manuscript, I understood that the authors confronted their data with data from Massip et al., Akerman et al. and Guilbaud et al. However, I am puzzled by the numbers of origins reported in the corresponding Supplementary Table3 that do not match the numbers in those publications:

- In, Massip et al., 155 395 origins are reported, where Table3 state 310,790.

- Number of all Core human origins from Akerman et al.: 64,000 core origins are reported and 127,248 in Table3.

- Iniseq2 origins from Guilbaud et al.: 23,905 reported origins Vs 47,634 in table3.

Thus, is this a mistake, misunderstanding? Or have the data been re-analysed with lowered thresholds, and how? I could not find the referred supplementary materials "p12: "(Fig. 7, supplementary tables 3 and 4, and supplementary materials)". But, if the data have been re-analysed, this should be clearly stated in the core manuscript and the figure itself, as this is not quite the same as only using published materials. Readers need to appreciate if those analyses have been carried in agreement/or not, and if so how, with the dataset/stringencies reported by the initial publications.

Thank you for allowing us to clarify this point that was confusing. The data have not been re-analyzed. However, in the legend of Table 3 was stated: "Associations were computed separately 1kb downstream (pG4 on the minus strand) and 1kb upstream the ori peak (pG4 on the plus strand), then combined". This means that we consider a given set of origins and perform the analyses twice (one analysis per orientation, since we cannot provide orientation for Oris). So for a given dataset (for

instance the Massip et al. data, 155,395), the analyses are based on twice the number of origins $155,395 * 2 = 310,790$. The important point here is that we did the same on the set of random sequences. Since the analysis framework is comparative (comparing the odds ratios), this artefactual data augmentation that combines (+) and (-) data has no impact on the final result.

For the Akerman data, the raw data contains 63,921 origins, but we removed the loci that correspond to chromosomes different from chr1 ... chrY, which makes 63,624 origins final. Same for the Ini-seq2 data : 23,824 initial origins and 23,817 final on canonical chromosomes.

The initial numbers are now indicated in the headers of the tables, and the caption was re-written to be more specific :

"Associations were computed separately 1kb downstream (pG4 on the minus strand) and 1kb upstream the ori peak (pG4 on the plus strand), then combined so the total number of origins corresponds to 2 times the initial number indicated in the table's header."

Minor points:

The authors should use "Iniseq2" wherever referring to Guilbaud et al. as I understood that Iniseq refers to Langley et al.

We thank the referee for his precision. We have changed Iniseq into Ini-seq2 in the main text and in legends.

Reviewer #2 (Remarks to the Author):

In this revised manuscript by the Prioleau's group, they have addressed in a satisfactory and appropriate manner the issues pointed by me and other referees. The findings that the presence of the two tandem G4-forming sequences (or clustered pG4s) is an important cis-element for active origins and that it is strongly associated with adjacent NFR add important novel information regarding the constituents required for vertebrate origin functions, although the findings point to many unanswered questions including what kind of higher-order structure are generated by the tandem pG4s at the origins, how they are generated and how they would facilitate origin firing, and how they are associated with efficient formation of NRF.

Other comments:

1. The authors state in the abstract that "Dimeric pG4s in replication origins trigger formation of an NFR next to precisely-positioned nucleosomes enriched in H2A.Z on this minimal origin and genome-wide."

However, they also state in Discussion that "It remains to understand how they might act. Do G4s promote the formation of NFRs or conversely would the presence of an NFR promote the formation of G4s?" Therefore, it is probably not accurate to state that "Dimeric pG4s in replication origins trigger formation of an NFR--". It would be more accurate to state at this stage that "Dimeric pG4s in replication origins are associated with formation of an NFR—".

We have amended the summary as suggested by the referee.

2. Figure 3D is helpful. However, the nomenclatures of constructs are still complicated. If authors can add schematic drawing of the constructs in the table (just to indicate the positions and orientations of pG4 and other features as well as the locations of the mutations introduced), it would be very helpful to the readers to visually grasp the data and conclusions.

We thank the referee for his comment. We have added a column containing schematic representations of the constructs. We agree with the referee that it will help the reader to follow the conclusions.

Reviewer #3 (Remarks to the Author):

In this paper, replication initiation activity was measured in several constructs containing guanine-rich motifs inserted into the beta-A globin locus in chicken DT40 cells and in another potential origin region at the Med14 promoter. The paper builds upon a previous publication (2014), which demonstrated a correlation between the formation of stable G-quadruplexes and replication origin activity. Additional constructs are reported, and it is shown that several combinations of G-quadruplex forming sequences at various intervals, and in particular, constructs that included two G-quadruplex forming sequences on the same strand, also facilitated replication origin activity. In line with previous studies, the insertion of these constructs was also associated with the formation of nucleosome free regions and some enrichment of histone H2AZ adjacent to replication origins. The paper also includes an analysis of genome-wide abundance of G-quadruplex forming sequences, utilizing primarily published datasets, demonstrating that clustered G-quadruplex forming sequences are enriched at replication origins and in the vicinity of “functional” elements, although they can be also present at non-origin regions.

This is a revised submission. The addition of several data points and some clarifications in view of the observations reported in the previous studies were useful. However, some of the concerns raised by the reviewers of the previous version remain unaddressed. Specific comments are listed below:

1. Abstract.

As currently written, the abstract does not explicitly state that the role of dimeric G-quadruplex forming sequences was investigated in a model system at select loci in chicken cells.

We have added “in avian DT40 cells” at the end of the following sentence: “We show that two pG4s on the same DNA strand (dimeric pG4s) are sufficient to induce the assembly of an efficient minimal replication origin without inducing transcription”.

Because it is becoming increasingly evident that replication origins are a diverse group and that replication initiation is oftentimes an indicator of chromatin accessibility, the association between specific chromatin features and replication initiation can be pertinent only to some select loci. The claim that the data “suggest a crucial role for dimeric pG4s in the organization and duplication of vertebrate genomes” requires further substantiation.

We agree with the comment of the referee that replication initiation is an indicator of chromatin accessibility. **We would like to emphasize that according to our supplementary figure 11B, 40% of NFRs are associated with an origin in chicken cells clearly showing that NFRs are not sufficient to**

make an origin. Moreover, only 12% of origins are associated with an NFR (Supplementary table 4). Therefore, what makes an initiation site in general is much more complex than just an accessible chromatin. The point of the paper is to show that the association of NFRs with clustered pG4s is a strong predictor for origin formation and not to explore all the potential mechanisms involved in the formation of efficient replication origins.

Regarding the “claim” that the data “suggest a crucial role for dimeric pG4s in the organization and duplication of vertebrate genomes”.

1) We made several constructs validated by two methods for their capacity to form an active and strong origin capable to locally advance the replication timing (and thus strong because active in the majority of the cells in the population) and showed that the necessary and sufficient *cis*-signal is two pG4s on the same strand.

2) We demonstrated on another strong origin (the med14 origin) the necessity of having two functional pG4s on the same strand to maintain this origin active and that the two necessary pG4s are sufficient to form an origin ectopically.

3) We showed genome-wide that ~50% (reaching 65% when several human cell lines are combined) of clustered pG4s are associated with a replication origin in human and chicken cells.

4) 38% of strong origins (top 25% most active origins in terms of SNS enrichment that provide ~ 80% of the SNS signal) contain a clustered pG4 in human (34% in chicken). **This implies that ~30% of the SNS signal is associated with clustered pG4s.**

5) 34% of strong origins mapped by Ini-seq2 contain a clustered pG4 in human cells. These origins are the first to fire in S-phase and therefore they shape the replication timing profile.

In conclusion, we strongly believe that our suggestion is appropriate and substantiated.

2. H2A.Z data. In agreement with the opinion of the reviewer #2, the conclusion that H2A.Z recruitment is controlled by pG4 density requires further substantiation.

We have changed the sentence: “Taken together, these genome-wide analyses suggest **that pG4s may participate in H2A.Z recruitment** but that this property is insufficient to define a functional origin.” By the following one “Taken together, these genome-wide analyses suggest that H2A.Z recruitment is insufficient to define a functional origin”(lanes 11-12, page 15).

3. Strong origins. The definition of strong origins at the top 25% of identified origins in terms of SNS enrichment is acceptable, however, it is unclear how G-quadruplex forming sequences can determine strong replication origins if they only associate with 16% of those origins.

The association of strong origins with clustered pG4 is not 16% (this number is for all origins) but 38% in human and 36% in chicken (Supplementary Tables 5 and 6). We realize that the information should be emphasized in the main text and therefore we added it (end of page 12).

How are other origins determined? Do G-quadruplex forming sequences play another role? These issues should at least be discussed.

Our study focuses on the role of dimeric pG4s in establishing efficient origins. We have not done anything on origins devoid of pG4 and therefore believe that a discussion of their regulation would be highly speculative and would therefore not provide any important information to the reader.

4. Origin efficiency. Related to the above, SNS enrichment of 30% is interpreted as strong origin activity in Figure 3 but as weak in Figure 1. This should be clarified.

We somehow already answered this question for the first revision. We were probably not clear enough.

We explain in the text why we use two assays for majoring origin functionality. “While quantification of SNS enrichment at the inserted β^A -globin promoter/origin assesses its capacity to induce replication initiation, measuring RT shifts can establish whether a change in initiation is occurring in the majority of a cell population, and also identifies an advance or delay in RT as positive or negative profile shifts, respectively”.

The replication timing assay, which is a functional assay, clearly demonstrates that the origin described in Figure 3 is efficient (Figure 3C). On the contrary, the RT assay applied to “weak origins” described in Figure 1 shows that they do not have the capacity to advance the RT locally. We agree with the referee that somehow, there is a discrepancy between the two assays and it is only the case for the origin described in Figure 3. We would like to point out that the standard deviation for SNS enrichment is stronger for this construct and that globally the SNS signal is higher than for the two weak origins described in figure 1. We already proposed a hypothesis in the main text: “The latter result confirmed that this origin is active and also suggested that the pattern of SNS enrichment observed probably reflected a more diffuse replication initiation site”. Another hypothesis is that the already described transitory block of the leading strand (Valton et al, 2014) is weaker with this combination of pG4s and therefore globally leads to a weaker SNS enrichment.

5. Structured G-quadruplexes. It would be good to define this term unambiguously and use consistently throughout the paper.

We agree with the referee and thank him for his suggestion. For ease of reading, we have written structured G4 each time we refer to them and no longer use the abbreviation G4.

6. Figure 1b. I was only able to see SNS tracks on the left-hand panel. It is crucial to show the SNS patterns for this construct, which forms the basis of the subsequent analyses.

Although we have made a whole genome analysis of the replication timing in a DT40 clone containing two copies of the β^A -globin full origin, we have not done a whole genome mapping of Short Nascent Strands in this cell line. This last analysis requires extensive work and is not necessary for the purpose of the paper. We respectfully disagree with the referee; the basis of the subsequent analyses is shown in Figure 1C. The relative SNS enrichment found at the ectopic β^A -globin full origin is analyzed in heterozygotes by qPCR (with two couples of primer pairs to be sure of the quantification and on two independent clones, grey bar). This is the most accurate way to precisely measure and compare the efficiency of different origins since there is no potential bias due to the construction of libraries and deep sequencing and it is feasible to analyze two independent clones with this method although it represents a considerable amount of work. Moreover, we need to compare all the mutants with exactly the same assays: SNS relative enrichment determined by qPCR and RT shift assay both made on heterozygotes and on several independent clones.

7. Figure 5a shows a very strong enrichment of histone H2A.Z in a region that is identified as a

nucleosome free region. How is this observation explained? Is there sufficient resolution to delineate the boundaries of the nucleosome free regions?

Nucleosome coverage and ChIP against anti-H2A.Z were performed on MNase digestions which generate mostly mono-nucleosomes (160 U as mentioned in the material and methods) shown in supplementary Figure 15. We made paired-end deep sequencing and in agreement with our protocol, most of the sequenced DNA is in the range of mono-nucleosomes. We therefore have the resolution to map the boundaries of NFRs as shown in figures 4, 5 and 8 and other supplementary figures. Regarding Figure 5A, to facilitate the understanding of the data, we added the track of ChIP H2A.Z coverage. We thank the referee for its suggestion since it provides a better delineation of the NFR region, between pG4#5 and pG4#3. The apparent strong enrichment of histone H2A.Z in the log2 ration H2A.Z/Input track is due to the low coverage of the input and probably to the smoothing used to obtain this profile. The new figure clearly shows the presence of an NFR flanked by strongly positioned nucleosomes containing H2A.Z.

8. Related to the above, the data suggest that nucleosome free regions are shown in both origin-associated and non-origin areas. How does this observation relate to the notion that NFRs determine origin activity? This observation should at least be discussed.

We partially answer in point 1, according to our supplementary figure 11B, 40% of NFR are associated with an origin in chicken cells clearly showing that NFRs are not sufficient to make an origin and therefore that what makes a strong initiation site is more complex.

The point of the paper is to show that the association of NFRs with clustered pG4s is a strong predictor for origin formation. To clarify this point we discuss this point in more detail p16, lanes 13-19.

9. Figure 9. As also pointed out by another reviewer in the previous review, the association between H2A.Z and origins is very modest and does not strongly support the figure title.

We have changed the title to: H2A.Z is significantly associated with replication origins and pG4s genome-wide, but not sufficient to induce the formation of an efficient origin.

Minor.

Figure 1c is cited in the text before figure 1b.

Figures 1a,b and c are all mentioned together at the beginning of the first paragraph in the result section.

“To delineate novel *cis*-motifs essential for metazoan replication origin activity within gene promoters, we employed a model origin construct that has the capacity to induce strong enrichment of short nascent strands (SNSs), a characteristic marker of replication origins, and to locally advance the replication timing (RT) of a middle-late replicated region in population-based assays in avian DT40 cell lines (**Fig. 1A–C**)”.

Figure 5A, the text is too small and it would be good to include more details in the legend.

We slightly increased the size of the typo and added more information in the legend.

REVIEWERS' COMMENTS

Reviewer #1 (Remarks to the Author):

The authors have clarified the points I raised in the last round of review, although I do not quite understand the following comments "we removed the loci that correspond to chromosomes different from chr1 ... chrY", but difference in term of origins analysed is negligible. However, some points raised by reviewer 3 in this last round, which are also criticisms raised by all three reviewers previously, remain unaddressed. I feel like the authors could have better acknowledged his/her comments in their revised version. My opinion remains the same since the first revision: The DT40 genetic experiments are interesting and well performed, the translation to genome wide by various bioinformatic analyses is informative (even though sometime difficult to follow), but the "crucial role for dimeric pG4" genome wide appear to me less obvious.

Reviewer #2 (Remarks to the Author):

Comments:

One of the crucial comments of the reviewer #3 was "The claim that the data "suggest a crucial role for dimeric pG4s in the organization and duplication of vertebrate genomes" requires further substantiation."

The authors' response explained detailed summary of the authors' data, and can be warranted. I believe that authors' suggestion is appropriate and substantiated.

Figure 1b, I agree with the authors that the results of SNS relative enrichment determined by qPCR would form the basis of the analyses, and a whole genome mapping of Short Nascent Strands would not be necessary for the purpose of the manuscript.

Other comments are also dealt with by the authors in a convincing manner, and I did not notice any technical concerns remaining.